



# Roles of marine biota in the formation of atmospheric bioaerosols, cloud condensation nuclei, and ice-nucleating particles over the North Pacific Ocean, Bering Sea, and Arctic Ocean

Kaori Kawana[1], Fumikazu Taketani[1], Kazuhiko Matsumoto[1], Takuma Miyakawa[1], Yutaka Tobo[2], Yoko Iwamoto[3], Akinori Ito[4], Yugo Kanaya[1]

[1] Earth Surface System Research Center, Research Institute for Global Change, Japan Agency for Marine-Earth Science and Technology (JAMSTEC), Yokohama, Kanagawa, 2360001, Japan
[2] National Institute of Polar Research, Tachikawa, Tokyo, 1908518, Japan
[3] Graduate School of Integrated Sciences for Life, Hiroshima University, Higashi Hiroshima, Hiroshima, 7390046, Japan
[4] Yokohama Institute for Earth Sciences, JAMSTEC, Yokohama, Kanagawa, 2360001, Japan

*Correspondence to*: Kaori Kawana (kawanak@jamstec.go.jp)

**Abstract.** We investigated the association of marine biological indicators (polysaccharides and protein-like gel particles, Chl-*a*) with the formation of fluorescent aerosol particles, cloud condensation nuclei (CCN), and ice-nucleating particles (INPs) over the North Pacific Ocean, Bering Sea, and Arctic Ocean during September–November 2019. The abundance of bioindicators was high in the North Pacific Ocean and the Bering Sea (e.g., up to 1.3 mg m$^{-3}$ of Chl-*a*), suggesting high biological activity known as the autumn bloom. In the North Pacific

Ocean, particles were characterized by high mass fractions of organics and sulfate with predominance of terrestrial air masses. Conversely, in the Bering Sea and the Arctic Ocean, particles were characterized by high mass fractions of sea salt and sulfate with predominance of maritime air masses. The averaged CCN concentration at 0.4 % supersaturation ranged from 99–151, 43–139, to 36 cm$^{-3}$ over the North Pacific Ocean with terrestrial influences, over the Bering Sea with marine biogenic influences, and over the Arctic Ocean with marine influences,

respectively, and the corresponding range of hygroscopicity parameter $\kappa$ was 0.17–0.60, 0.42–0.68, and 0.67, respectively. The averaged INP concentration ($N_{INP}$) measured at temperatures of −18 and −24 °C with marine sources was 0.01–0.09 and 0.1–2 L$^{-1}$, respectively, and that over the Arctic Ocean was 0.001–0.016 and 0.012–0.27 L$^{-1}$, respectively. When marine sources were dominant, fluorescent bioaerosols were strongly correlated with all bioindicator types (*R*: 0.81–0.88) when considering the wind-uplifting effect from the sea surface to the

atmosphere. Correlations between $N_{INP}$ measured at −18 and −24 °C and all bioindicator types (*R*: 0.58–0.95 and 0.79–0.93, respectively) and between $N_{INP}$ and fluorescent bioaerosols (*R*: 0.50 and 0.60, respectively) were also





positive, suggesting that marine bioindicators contributed substantially as sources of bioaerosols and cloud formation.

## 1. Introduction

Aerosol particles can affect the climate system directly by absorbing and scattering solar radiation and indirectly by radiative cooling and albedo modulation via cloud processes. Despite progress, process-level understanding of aerosols remains far from complete, particularly owing to their diverse roles in aerosol–cloud interactions and complex links to natural systems that include the oceans (IPCC AR6, 2021). Primary biological aerosol particles, originating from sea surface organic matter (OM) derived from marine ecosystems, could be uplifted into the atmosphere as sea spray aerosols (SSAs) (Cochran et al., 2017). Such particles could affect cloud processes by acting as cloud condensation nuclei (CCN) and ice nucleating particles (INPs). Biological aerosol particles (bioaerosols) are composed of biological material, such as bacteria, algae, fungal spores, pollen, leaves, and cell pigments, and they might be detectable by their fluorescence (Després et al., 2012; Fröhlich-Nowoisky et al., 2016; Fennelly et al., 2018). Online methods, Wideband Integrated Bioaerosol Sensors (WIBS), and Ultraviolet Aerodynamic Particle Sizers (UV-APS) have all been applied to specific target species such as amino acids, proteins, and coenzymes in cells (Pöhlker et al., 2012; Gabey et al., 2010). More visually, microscopic or imaging methods have been adapted to specifically stain the cell nucleus and DNA/RNA for living cell counts (Maki et al., 2013; Fröhlich-Nowoisky et al., 2016). As potentially related sea surface OM derived from marine ecosystems, studies have investigated gel-like organic particles, such as polysaccharide-containing transparent exopolymer particle (TEP and protein-containing Coomassie stainable particle (CSP), because they aggregate in the microlayer under low wind speed (WS) conditions and are released into the atmosphere via wave breaking under high WS conditions (Gantt et al., 2013; Sun et al., 2018).

Bioaerosols originating from biogenic OM in marine ecosystems can contribute to both CCN and INPs during periods of high biological activity (Vergara-Temprado et al., 2017; Creamean et al., 2019). The interaction and partitioning between CCN and INPs originating from primary biological aerosol particles with water saturation in mixed-phase clouds are important for understanding the abundance of cloud droplets and ice crystals in cloud processes (Quinn and Bates, 2014; Brooks and Thornton, 2017, Burrows et al., 2022). Generally, CCN and INPs have different activation characteristics in terms of chemical composition and size. For example, mainly highly hygroscopic particles with high solubility such as sulfate and sea salt in submicron particles are likely to be active as CCN, whereas less hygroscopic particles such as dust and volcanic ash in the coarse mode are likely to be active



as INPs. For CCN activation, local biogenic OM and organic aerosols, including surfactants, act as CCN (Facchini et al., 2008; Fuentes et al., 2011; Ovadnevaite et al., 2011) mainly in submicron particles, while more hygroscopic components such as sulfate and sea salt also contribute largely in super-micron particles (Gong et al., 2019). Some CCN grow to giant CCN and activate as cloud droplets of larger size (Sun and Ariya, 2006; Möhler et al., 2007).

For INP activation, mineral dust particles are the primary contributors to INP concentrations with long-range transport, especially in the Northern Hemisphere (DeMott et al., 2010, 2015), but biomaterials (i.e., bacteria and algae) might contribute largely in low-level and mixed-phase clouds in regions without influence from dust/terrestrial sources (Burrows et al., 2013) because they can activate as INPs at relatively high temperatures, i.e., higher than –10 °C (Hoose and Möhler, 2012). INP activation of biogenic materials is effective in comparison with

the case of homogeneous ice nucleation (−36 °C) or other typical materials such as dust particles (Murray et al., 2012). However, in comparison with terrestrial bioaerosols (e.g., OM in forests and soil), the amount, behavior, and appropriate indicators of marine bioaerosols remain poorly understood owing to lack of observations. Therefore, information gained from in situ observations and modeling studies on the production of bioaerosols and their emission from the sea surface to the atmosphere, their abundance in the atmosphere, and their relative relationships

with CCN and INPs, is critical when considering the importance of bioaerosols to the radiative budget and climate effects via aerosol–cloud processes. Previous comprehensive observational studies focusing on sea surface–aerosol–cloud interactions are limited (Abbatt et al., 2019; van Pinxteren et al., 2020; McFarquhar et al., 2021), and only a few studies have discussed the response of the abundance of CCN and INP number concentrations and activation properties to biogeochemical and environmental conditions based on in situ observations in the Southern

Ocean and Antarctic (Schmale et al., 2019; Tatzelt et al., 2021), Arctic Ocean (Shupe et al., 2022), and Mediterranean Sea (Gong et al., 2019). Recently, an annual cycle of observation was conducted during the Multidisciplinary drifting Observatory for the Study of Arctic Climate (MOSAiC) expedition to investigate the relationship between aerosols and clouds, and to assess the impact of aerosols on the climate. Preliminary results revealed seasonal variation in the number concentrations of CCN and INPs relative to that of total particles (Shupe

et al., 2022). However, the specific relationships between biogenic OM in seawater, fluorescent particles that are proxies for bioaerosols, and cloud particle formation remain unclear, and comprehensive cross-sectional studies have yet to be conducted.

In this study, shipboard observations were obtained in the North Pacific Ocean, Bering Sea, and Arctic Ocean to investigate the biogeochemical processes leading to marine biogenic sources, fluorescent bioaerosols, and CCN

and INPs between the sea surface and the atmosphere. We present the physicochemical properties of aerosol particles and oceanic precursors (i.e., chemical composition, number, and mass concentrations) and discuss the



parameters of CCN and INP activation in relation to the biogeochemical cycle between the atmosphere and ocean. Moreover, we provide equations to link the amount of marine bioaerosols, CCN, and INP concentrations from marine bioindicators with correlation analysis, for assessment of the impact of biological indicators on marine fluorescent bioaerosols and CCN and INP activation.

## 2. Cruise observations

### 2.1. Online measurements of atmospheric aerosols and trace gases

Cruise observations were conducted in the North Pacific Ocean, Bering Sea, and Arctic Ocean onboard R/V *Mirai* (JAMSTEC) during 27 September to 10 November 2019. For online measurement of atmospheric aerosol
particles, an inlet for total suspended particles (model: URG-2000-30DG, URG, NC, USA) was placed approximately 18 m above sea level on the compass deck. A WIBS (model -4A, Droplet Measurement Technologies, Longmont, CO, USA) was placed on the compass deck and particles were introduced directly via the inlet to the instruments at ambient relative humidity. Other instruments, namely, a scanning mobility particle sizer (SMPS; model: 3910, TSI, USA), a CCN counter (CCNC; model: 100, Droplet Measurement Technologies,
Longmont, CO, USA), and a mixing condensation particle counter (MCPC; model: 1720, Brechtel, Hayward, CA, USA), were installed in the research laboratory room. Particles fed via the inlet were dried in a Nafion™ tubing drier (model: MD-700, Perma Pure Inc., NJ, USA) to maintain the relative humidity below 40 %, and the particles were introduced to the instruments via conductive tubes.

Fluorescent aerosol particles (FAPs) were observed by the WIBS-4A, which detected autofluorescence from
individual particles in two wavelength bands (310–400 and 420–650 nm) upon excitation using two xenon flash lamps emitting at different wavelengths, i.e., 280 and 370 nm (Healy et al., 2012). The FAPs were classified into seven types depending on fluorescence pattern: fluorescence in a single wavelength as Type A (excitation = 280 nm, fluorescent = 310–400 nm), Type B (excitation = 280 nm, fluorescent = 420–650 nm), or Type C (excitation = 370 nm, fluorescent = 420–650 nm), and a combination of fluorescence as Types AB, AC, BC, or ABC (Perring
et al., 2015). Size distributions of fluorescent and nonfluorescent particles were derived from the scattering intensity measurements obtained using a continuous-wave 635 nm diode laser (O'Connor et al., 2013). Previous studies reported that either a 3σ or 9σ signal level could be used as a baseline threshold to distinguish fluorescent particles from others, where the 1σ level was determined as the background fluctuation in the absence of particles (Crawford et al., 2016). In this study, we used 3σ as the threshold value. Fluorescent polystyrene latex particles with size of 2



µm (PSL, G0200, Thermo Fisher Scientific, Waltham, MA, USA) were introduced before and after the observations were conducted to check the validity of the particle size and fluorescent intensity detected by the instrument (Robinson et al., 2017; Savage et al., 2017).

The number–size distributions and number concentrations of aerosol particles from 10 to 415.8 nm in terms of

the mobility particle diameter ($D_{mob}$) under dry conditions were observed by the SMPS at 1 min intervals. The CCN number concentrations ($N_{CCN}$) and condensation nuclei (CN) number concentrations were observed by the CCNC and the MCPC, respectively. Ammonium sulfate particles (purity: 99.999%, Sigma-Aldrich, USA) were introduced to the CCNC before and after the observations were conducted to confirm the supersaturation (SS) conditions at the setting of 0.1, 0.2, 0.4, and 0.7 %. The SS conditions of the CCNC were switched every 15 min (one cycle of 1

h). In the analysis of CCN, the data points in the stable flow rate (ratio of sample to sheath flow of 9.5–10.5) were only used, and the data points of the final 3 min were used for analysis under stable SS conditions. The CCN activation diameter ($d_{act}$) was calculated as the critical diameter at which $N_{CCN}$ was equal to the total particles ($N_{total}$, 10–415.8 nm) with number–size distributions derived from the SMPS in parallel, under the assumption that particles activate as CCN from larger particles with an internal mixing state. Then, the hygroscopicity parameter $\kappa$

was derived from $d_{act}$ based on the $\kappa$-Köhler equation (Petters and Kreidenweis, 2007):

$$S = \frac{d_{p,\,wet}{}^3 - d_{p,\,dry}{}^3}{d_{p,\,wet}{}^3 - d_{p,\,dry}{}^3(1-\kappa)} \exp\left(\frac{4\sigma_s M_w}{RT\rho_w d_{p,\,wet}}\right) \qquad (1),$$

where $S$ is the equilibrium supersaturation, $\sigma_s$ is surface tension, $M_w$ is the molecular weight of water, $R$ is the universal gas constant, $T$ is absolute temperature, and $\rho_w$ is the density of water. Here, $T$ was assumed to be 300 K, i.e., the mean temperature of the top of the CCNC column. The surface tension of $\sigma_s$ value was assumed to be that

of pure water (71.5 mN m$^{-1}$). The detailed procedure of the calculation is described in Kawana et al. (2016). The activation fraction (AF) was obtained as the ratio of $N_{CCN}$ in the $N_{total}$. In the trace gas measurements, ozone (O$_3$) and carbon monoxide (CO) concentrations were measured using UV (model: 205, 2B Technologies, Boulder, CO, USA) and nondispersive infrared (model: 48C, Thermo Fisher Scientific, Waltham, MA, USA) sensors. The data points were removed when large instabilities or reduced levels in the 1 min O$_3$ were recorded, indicating titration

by NO in the fresh exhaust (Kanaya et al., 2019). All data points from the online measurements were screened using the same criteria that were applied to the operation of the pump of the high-volume air sampler to avoid contamination from ship exhaust (See Section 2.2.).

**2.2. Offline measurements of atmospheric aerosols and seawater sampling**



In the offline measurements, a high-volume air sampler (model: HV-525PM, Sibata Scientific Technology, Ltd., Tokyo, Japan) used for chemical analysis was placed on the compass deck, and particles with diameter of <2.5 μm (PM$_{2.5}$) were collected on a quartz filter (model: QR-100, Advantec, Tokyo, Japan) using an impactor at a flow rate of approximately 500 L min$^{-1}$ every 2 d. Additionally, a custom-made air sampler for INP analysis was installed

and non-size-selective ambient particles were collected onto a polycarbonate membrane filter (0.2 μm pores, Whatman) at a flow rate of 10 L min$^{-1}$ every 2 d. To prevent contamination from ship exhaust, a wind selector was used to stop the sampling pumps automatically when the wind direction deviated by more than ±75° from the bow direction or when the WS fell below 2 m s$^{-1}$. Samples for analysis of INPs and chemical composition were collected in a sterile centrifuge tube and then stored prior to analysis at approximately 4 °C in a refrigerator and −20 °C in a

freezer, respectively. The mass concentrations of ionic species (NH$_4^+$, Na$^+$, K$^+$, Ca$^{2+}$, Mg$^{2+}$, Cl$^-$, NO$_3^-$, and SO$_4^{2-}$) were measured by ion chromatography (model: ICS-1000, Dionex Co., CA, USA), and the mass concentrations of sea salt and non-sea-salt sulfate (nss-SO$_4^{2-}$) were calculated from Na$^+$ using an equation of standard seawater (Warneck, 2000). The mass concentrations of organic carbon (OC) and elemental carbon (EC) in the PM$_{2.5}$ were also obtained using a thermal/optical carbon analyzer (model: DRI 2001, Desert Research Institute, Reno, NV,

USA) with the Interagency Monitoring of Protected Visual Environments protocol, and the mass concentrations of water-insoluble organic carbon were derived by subtraction of the measured water-soluble organic carbon from the total OC. Levoglucosan was analyzed using a derivatization gas chromatography mass spectrometer (model: GCMS-QP2010Plus, Shimadzu Co., Kyoto, Japan). The number concentrations of INPs upon immersion freezing were obtained using the National Institute of Polar Research Cryogenic Refrigerator Applied to Freezing Test

(Tobo, 2016). From the detections made between 0 and −30 °C with a 0.5 °C step, the number densities of INPs determined at three selected temperatures (−18, −24, and −30 °C) were used for analysis in this study. The detailed extraction and analysis procedures for INP measurements are described elsewhere (Tobo, 2016; Tobo et al., 2020).

Surface seawater sampling for TEPs and CSPs was performed using a bucket at 22 sampling stations. In analysis of both TEPs and CSPs, 200 mL of seawater was filtered through a Nuclepore™ polycarbonate membrane filter

(cut size: 0.4 μm, Cytiva, Tokyo, Japan) and triplicate filters were obtained from each seawater sample. For TEPs, 1 mL of Alcian blue staining solution, adjusted to pH 2.5, was added to the filter and the filter was rinsed three times with 1 mL of Milli-Q® water after 4 s of staining. Filters were soaked for 2 h in 6 mL of 80 % sulfuric acid for extraction and absorbance was measured at the wavelength of 787 nm (Alldredge and Passow, 1993). For CSPs, 1 mL Coomassie brilliant blue staining solution was added to the filter, which was then rinsed five times with 1

mL of Milli-Q® water after 1 min. Filter samples were soaked for 2 h in 4 mL of 3 % sodium dodecyl sulfate in 50 % isopropyl alcohol with ultrasonic extraction to elute the dye, and the absorbance of the solution was measured



at the wavelength of 615 nm (Cisternas-Novoa et al., 2015). The calibration curves for the relationship between absorbance/weight and abundance for TEPs and CSPs were produced using a xanthan gum solution (XG, Sigma-Aldrich Co. LLC, St. Louis, MO, USA) and bovine serum albumin (BSA, Sigma-Aldrich Co. LLC, St. Louis, MO, USA) as a standard solution, respectively. Then, the TEP and CSP concentrations were quantified as XG equivalent (Passow and Alldredge, 1995) and BSA equivalent (Thornton, 2018) concentrations, respectively. The concentrations of Chl-*a* and nutrients in the surface seawater were determined using a fluorometer (model: 10-AU, Turner Designs, Inc., San Jose, USA) and a continuous segmented flow analyzer (model: QuAAtro 2-HR, BL TEC K.K., Tokyo, Japan), respectively. Meteorological parameters at the sea surface, such as WS and sea surface temperature, were measured by the monitoring system onboard R/V *Mirai*.

## 3. Results and discussion

### 3.1 Air mass origin, chemical composition, and trace gases

We identified five periods with different air mass origins in different sea regions during the cruise (Fig. 1a–e): Period 1 (P1, 30 September to 6 October 2019, in the North Pacific Ocean), where air masses originated from terrestrial regions with anthropogenic sources (i.e., the Asian continent); Period 2 (P2, 7–9 October 2019), Period 3 (P3, 11–27 October 2019), and Period 4 (P4, 28 October to 2 November 2019) with potential influences from the Bering Sea, Arctic Ocean, and Bering Sea, respectively, and Period 5 (P5, 3–7 November 2019, in the North Pacific Ocean), where influences from anthropogenic sources were as evident as in P1. Here, air mass origin was classified according to 5-d backward trajectories using the NOAA HYSPLIT model (Stein et al., 2015) from a starting altitude of 500 m, based on the Global Data Assimilation System with $1\,° × 1\,°$ resolution from the National Center for Environmental Prediction. The characteristics of observational parameters in each period are summarized in Table 1.

The temporal variations in the mass concentrations of major components and mass fractions of chemical components (OC, EC, inorganics ($NH_4^+$, $NO_3^-$, and $SO_4^{2-}$), and sea salt) in the $PM_{2.5}$ particles are shown in Fig. 2a–e. The compositional characteristics show clear response to changes in air mass; OC and sulfate represented the main components during P1 and P5 (15–22 % and 28–48 %, respectively), implying strong influence from terrestrial sources. Sea salt was the main component during P2 and P3 (76–88 %), and the plots of mass concentrations of OM and $Na^+$ were correlated positively (*R*: 0.66), suggesting predominance of local marine sources including biogenic OM with SSAs. During P4, OC, sulfate, and sea salt were dominant (15, 31, and 43 %, respectively). During P5 when the air masses originated from the Asian continent, the mass concentrations of $Ca^{2+}$



and $Mg^{2+}$, as indicators of dust mineral particles, and those of levoglucosan, as an indicator of biomass burning, had the highest values (averages of 0.07 µg m$^{-3}$ for $Ca^{2+}$ and $Mg^{2+}$ and 8.1 ng m$^{-3}$ for levoglucosan (Fig. 2d)), whereas the levels of those indicators were lower in the other periods, implying that the influence of both mineral dust particles and biomass burning was limited. The $O_3$ and CO concentrations also varied in accordance with the

air mass classification; they were high in the North Pacific Ocean (~42 and ~120 ppb, respectively) and low in the Arctic Ocean and the Bering Sea (35–38 and ~110 ppb, respectively) (Fig. 2f and 2g).

### 3.2 Distributions of biological indicators in surface seawater

Temporal variations of Chl-$a$ and biological organic gel particles (TEPs and CSPs) in the surface seawater are
shown in Fig. 3a and 3b, and those of nutrients are shown in Fig. S1. The spatial distributions (Fig. 3c and 3d) show that the concentrations of TEPs, CSPs, and Chl-$a$ were relatively high in the North Pacific Ocean and the Bering Sea (mean values ± one standard deviation of 73 ± 34 µg XGeq L$^{-1}$, 24 ± 22 µg BSAeq L$^{-1}$, and 0.86 ± 0.23 mg m$^{-3}$, respectively), coincident with high nutrient contents and suggesting high biological activity (e.g., an autumn bloom). Particularly, high concentrations of bioindicators (i.e., TEPs, CSPs, and Chl-$a$) were observed from the
Bering Sea to the Chukchi Sea (P2 and P4, Fig. 3a–d), corresponding to changes in nutrient concentrations. Conversely, over the Arctic Ocean, the concentration of CSPs decreased markedly to 12 ± 13 µg BSAeq L$^{-1}$, while TEPs and Chl-$a$ maintained relatively high concentrations (47 ± 10 µg XGeq L$^{-1}$ and 0.33 ± 0.12 mg m$^{-3}$, respectively) in comparison with those previously reported during summer (August–September) (Park et al., 2019, TEPs: ~20 µg XGeq L$^{-1}$, CSPs: ~20 µg BSAeq L$^{-1}$, and Chl-$a$: ~0.2 mg m$^{-3}$).

In this area, blooms have been observed from late summer to autumn in previous studies. Autumn blooms might be triggered by wind-induced mixing in biological hotspots, where nutrient-rich water originating from the Bering Sea flows into the Chukchi Sea and produces high concentrations of Chl-$a$ and biological OM (Nishino et al., 2016). Matsuno et al. (2015) also reported that the supply of nutrient-rich water from the deep layer can lead to phytoplankton blooms with an increase in large-sized Chl-$a$ (~10 µm) and variation in the zooplankton community
in the biological hotspot of this area. Our results indicate that high concentrations of marine primary OM such as TEPs and CSPs might have resulted from high biological activity due to an autumn bloom associated with the mixing of nutrient-rich water. As mentioned in Sect. 3.1, backward trajectory analysis suggests that air masses with marine sources were generally free of terrestrial/anthropogenic influence (Fig. 1c) during this period, and that sea salt was the major component with OM accounting for 5–10 % of the mass of PM$_{2.5}$ (Fig. 2e). This indicates that
primary organic aerosols from biological activity in the ocean surface might have been ejected into the atmosphere



as SSAs associated with wave breaking under high WS conditions (Sun et al., 2018). This process might have been strongly involved in the formation of marine bioaerosols, which could be relevant to local major sources of CCN and INPs in remote open ocean areas. We discuss the links between biological activity on the sea surface and bioaerosols in the atmosphere in Sect. 3.3, and examine the consequent impacts on cloud processes in the Arctic in
Sect. 3.4 and Sect. 3.5.

## 3.3 Fluorescent aerosol particles (FAPs) over the North Pacific Ocean, Bering Sea, and Arctic Ocean

Figure 4a shows the time series of the number concentrations of FAPs and total (fluorescent and nonfluorescent)
particles with the optical diameter ($D_p$) of >1 μm (i.e., $1 < D_p < 2.5$ μm) measured by the WIBS-4A. The average number concentration of FAPs during the entire observation period was 25 particles $L^{-1}$, representing a 2.5 % fraction of the total. In terms of the classified fractions of particles based on seven fluorescence patterns (Fig. 4b), Types A (41 %), B (28 %), and AB (10 %) were dominant. The total number density of FAPs was correlated positively with WS during the entire observation period ($R$: 0.42) (Fig. 4c and 4d). The correlation between FAPs
and WS was high during P2 ($R$: 0.57) and P4 ($R$: 0.80) when the marine biogenic source was more dominant than in other periods, suggesting that the FAPs over the ocean originated from local emission from the sea surface in association with strong winds and high waves. The number fraction of Type C particles increased over the Arctic Ocean (P3), while that of Types B and BC increased near land at the end of the period (P5). In the case of coarse mode particles ($D_p > 2.5$ μm, Fig. S3), the average number concentration of FAPs was 7 particles $L^{-1}$ and the
fraction was 7 %. The fraction of particles emitting fluorescence in multiple-band types such as Type AB (17 %) and Type ABC (14 %) increased, while that in Type A (28%) and Type B (31%) was also dominant. This result is consistent with observations of primary biological aerosol particles emitted with SSAs during summer in the pristine Southern Ocean, i.e., Types A, B, AB, and ABC were the major patterns, and fluorescent particle concentrations ranged from $10^{-1}$ to $10^{1}$ particles $L^{-1}$ (Moallemi et al. 2021). Regarding the variation of fluorescence
pattern, a previous study based on laboratory experiments using certain bioaerosol materials (i.e., bacteria, fungi, spores, and pollen) found that Type A particles originated from bacterial and fungal species (Hernandez et al., 2016). Our previous oceanographic observations over the central Pacific (Kawana et al., 2021) similarly showed that FAPs in Type A and Type C predominated (75 %) in clean remote oceanic air masses, and that their abundance correlated well with oceanic TEPs (polysaccharide polymer) and bacteria, when considering the influence of WS
in the formation of SSAs, while FAPs in Type B were dominant (30 %) near land and strongly correlated with



CSPs (protein-like polymers). The identity of marine bioaerosols detected by fluorescence observations was certified by comparison to a DNA staining method. Santander et al. (2021) also suggested the dominance of bacteria as Type A and Type C in the FAPs identified by a WIBS and the three-dimensional excitation–emission matrix fluorescence spectra during Marine Aerosol Reference Tank experiments using sampled seawater (during

bloom/non-bloom periods) or cultured seawater. These results indicate that the dominance of Type A and Type B might reflect different sources/origins and contribute to forming bioaerosols in the marine environment.

To further focus on the relationship between marine biota and marine bioaerosols in this study, we extracted data only for P2, P3, and P4 when marine influences were dominant. We assessed the relationship between bioindicators in seawater and the amount of marine biological aerosols over the ocean; that is, to establish how well

the abundance of fluorescent particles could be predicted by the amount of oceanic bioindicators and their uplift into the atmosphere, and to determine which would be the most relevant bioindicator (Fig. 5, Table 2). The method, used in our previous study characterizing atmospheric bioaerosols over the central Pacific Ocean (Kawana et al., 2021), confirmed that the ocean surface abundance of TEPs and bacteria showed strong correlation ($R > 0.80$) when multiplied by WS, indicative of the origin of marine bioaerosols in remote ocean areas. In this study, we examined

the correlation for both fine ($1 < D_p < 2.5$ μm) and coarse ($D_p > 2.5$ μm) particles, and we found a tendency for higher correlation coefficients for the fine particles (Table 2). This might be partly because larger particles are less likely to be uplifted into the atmosphere and are susceptible to complex deposition processes. The correlation coefficient between FAPs and WS in the maritime airmass ($n = 13$) was medium ($R$: 0.58). The FAPs showed even weaker correlation with Chl-$a$ ($R$: 0.33) and with TEPs or CSPs ($R < 0.1$). Nonetheless, the correlation became

significantly improved when the local WS was multiplied by the quantities of all bioindicators ($R$: 0.81–0.88). When the wind effect was considered as the square of WS, high correlation coefficients were also obtained ($R$: 0.88–0.93). Regarding the order of WS dependence, it has been reported that atmospheric SSA concentrations can be represented by exponents from 0.68 to 2.8 power of the WS at the sea surface (Jaeglé et al., 2011; Ovadnevaite et al., 2012); the correlation coefficients were almost unchanged when an exponent of 2 was used instead of 1 (see

Table 2). These results strongly indicate that marine biota contributed to bioaerosol formation. From linear regression, we propose the following equations to predict the abundance of marine bioaerosols (Fig. 5e–g):

[bioaerosols] (particles L$^{-1}$) = (0.045 ± 0.007) · [TEPs] (μg XGeq L$^{-1}$) · WS (m s$^{-1}$) + (0.63 ± 2.8) ($R$: 0.88),  (2)

[bioaerosols] (particles L$^{-1}$) = (0.084 ± 0.019) · [CSPs] μg BSAeq L$^{-1}$) · WS (m s$^{-1}$) + (8.8 ± 2.2) ($R$: 0.81),  (3)

[bioaerosols] (particles L$^{-1}$) = (2.6 ± 0.48) · [Chl-$a$] (mg m$^{-3}$) · WS (m s$^{-1}$) + (7.7 ± 2.2) ($R$: 0.85).  (4)

Note that lower correlations (~0.5) were obtained when we considered nonfluorescent particles (Fig. S4), indicating that fluorescent particles on the ocean surface particularly contribute to the formation of bioaerosols. The





correlation coefficients are similar but the slope of the regression line for TEPs is 40% smaller than the case over the central Pacific Ocean (slope: $0.076 \pm 0.014$, $R$: 0.88, Kawana et al., 2021), suggesting the possibility of dependence on phytoplankton communities in the different oceanic regions. Among the marine indicators, polysaccharides (TEPs) were reported as a major group of ocean-derived OM in seawater samples and SSAs during

the ICEALOT cruise, and they might represent an important contributor to CCN in the Arctic Ocean (Russell et al., 2010). Park et al. (2019) also found that TEPs correlated well with the number concentration of SSA particles ($R$: 0.86–0.99) over the Arctic Ocean during summer; however, their analysis did not include bioaerosols. These results support the assertion that TEPs (with bacteria, in some cases) represent a major contributor to marine bioaerosols over the ocean. Furthermore, it is unique to find Chl-$a$ as a good predictor (Eq. 4) of marine bioaerosols over high-

latitude regions, in contrast to our previous result over the central Pacific Ocean that showed poor correlation (slope: $20.0 \pm 19.0$, $R$: 0.47). Nonetheless, the slopes for the two cases agree within the range of uncertainty. The good correlation of fluorescent bioaerosols and Chl-$a$ in this study might partly be attributable to the strong correlation between Chl-$a$ and TEPs or CSPs ($R$: 0.64–0.67); the TEP/Chl-$a$ ratio might vary depending on the key phytoplankton community (Engel et al., 2017). Future studies over different oceanic regions are required to

comprehensively assess the association of Chl-$a$ in bioaerosol formation because Chl-$a$, which is available from in situ and remote observations (i.e., satellite methods), is useful for input in model calculations as biological activity.

While we obtained good correlations and predictive equations for the abundance of bioaerosols from all bioindicators when we focused on marine primary organic aerosols in the fine mode, other factors should be investigated to derive accurate equations for linking bioindicators in seawater to bioaerosols in the atmospheric

aerosols. Previous studies suggested that remaining issues that might cause discrepancies comprise the following: (1) the dependence of organic enrichment on biogenic OM types in the aerosol particles (i.e., functional group as polysaccharide, protein, lipid, and sugar) corresponding to the phytoplankton community and bloom stage (Cravigan et al., 2020; McCluskey et al., 2018a; Ickes et al., 2020; Moallemi et al., 2021) and their description in models (Burrows et al., 2014), (2) selective transfer of OM (i.e., humic-like, protein-like, and fluorescence pattern)

in the seawater to aerosol phase (Rastelli et al., 2017; Miyazaki et al., 2018; Jung et al., 2020; Santander et al., 2021, 2022), and (3) delay response time to form bioaerosols between atmospheric aerosols (both primary and secondary organic aerosols) and bioindicators in the sea surface (O'Dowd et al., 2015; Freney et al., 2021; Sanchez et al., 2021). Thus, detailed analysis of the behavior of biological particles with OM type and response to cloud activation is desirable in future study.





### 3.4. Physicochemical properties of aerosol particles, CCN, and INPs

The averaged number–size distributions during each period ($D_{mob}$: 10–415.8 nm) are presented in Fig. 6. The averaged aerosol number concentrations during P1, P2, P3, P4, and P5 were $231 \pm 144$, $63 \pm 30$, $80 \pm 50$, $184 \pm 72$, and $356 \pm 238$ cm$^{-3}$, respectively. In the North Pacific Ocean (P1 and P5), the observed aerosol number

concentrations were higher than those in the other periods, with particular dominance of fine particles in the Aitken mode. Both $O_3$ and EC, indicators of pollutants from anthropogenic activity, also showed higher concentrations than in other periods (Fig. 2), suggesting influence of terrestrial/anthropogenic particles from nearby land areas and the Asian continent. This is also confirmed by the backward trajectory analysis shown in Fig. 1. In the Bering Sea and the Arctic Ocean (P2, P3, and P4), when particles from local marine sources were dominant, lower aerosol

number concentrations were observed, exhibiting a bimodal distribution with a major peak in the accumulation mode. This characteristic is common in clean marine environments where sulfate and sea salt particles in the accumulation and coarse mode are dominant (Lange et al., 2018). These trends (i.e., mode diameter, number concentration, and number-size distribution) are consistent with those found in previous studies over the Arctic Ocean and the North Pacific Ocean during summer (August–September), with reported average number

concentrations of particles from 3–300 nm of 1622, 413, and 397 cm$^{-3}$ for Arctic terrestrial, Arctic marine, and Pacific marine air masses, respectively, and the cause of the high concentrations was interpreted as frequent occurrence of new particle formation (Park et al., 2020). In our study, the aerosol number concentration during P5 in the North Pacific was almost the same, whereas the values during other the periods were lower. We did not observe new particle formation events (i.e., the so-called "banana-shaped" growth in particle size); instead, we

observed background conditions over the remote ocean in a later season than that reported by Park et al. (2020).

For CCN activation, the time series of aerosols and CCN number concentrations, CCN activation diameter ($d_{act}$), and hygroscopicity parameter $\kappa$ are presented in Fig. 7a and 7b, and the AF as a function of SS is presented in Fig. 7c. The averaged $N_{CCN}$ values at 0.4 % SS during P1, P2, P3, P4, and P5 were $99 \pm 44$, $43 \pm 22$, $36 \pm 13$, $139 \pm 44$, and $151 \pm 85$ cm$^{-3}$, respectively. The $N_{CCN}$ value was low despite the high concentration of aerosols during P1,

especially near land, i.e., approximately 50 % of aerosol particles were activated as CCN at 0.4 % SS. This can be attributed to the effects of both particle size and chemical composition. For example, smaller particles require high SS conditions to be activated as CCN, and they are typically enriched in less hygroscopic components in the Aitken mode. Indeed, the $\kappa$ values at 0.4 % SS derived from $d_{act}$ during P1 were particularly low (mean value: $0.17 \pm 0.06$, Table 1) when the influence of terrestrial pollutants was large, suggesting a notable contribution of less hygroscopic

components such as OM and EC from terrestrial/anthropogenic sources. During P2, the $\kappa$ values were also much


lower (mean: $0.42 \pm 0.04$) than those for sea salt ($\kappa \sim 1.2$), although a high mass fraction of sea salt was observed in the $PM_{2.5}$ particles (88 %). In this period, high concentrations of marine bioindicators were observed (see Sect. 3.2 and Sect. 3.3), i.e., gel organic particles (e.g., TEPs and CSPs) with lower solubility and less hygroscopicity. These OM types, together with sea salt, are suggested to contribute substantially to the reduction in particle

hygroscopicity, especially for fine particles around the CCN activation size range. Additionally, some types of water soluble OM ($\kappa \sim 0.3$, Petter and Kreidenweis, 2007) such as dicarboxylic acid (i.e., oxalic acid) as secondary biogenic OM and nss-sulfate from high biological activity might also have contributed (Kawamura, 2023). In contrast, during P3 (in the Arctic Ocean) and P4, the CCN activation parameters were characterized by high $\kappa$ values (0.67–0.68) and high CCN AFs (64–76 %), indicating that hygroscopicity and CCN activity were controlled

by the highly hygroscopic components from natural sources such as sea salt ($\kappa \sim 1.2$) and sulfate ($\kappa \sim 0.6$). The $\kappa$ value during P5 was approximately equivalent to that of ammonium sulfate ($\kappa \sim 0.6$), and the influence from the terrestrial region might have been larger, as indicated by the backward trajectories. The averaged $N_{CCN}$ values at 0.4 % SS (36–139 $cm^{-3}$ during P2, P3, and P4 and 99–151 $cm^{-3}$ in the North Pacific Ocean) were similar to the $N_{CCN}$ values measured in the remote Arctic Ocean ($\sim$50–70 $cm^{-3}$, Jung et al., 2018; Lange et al., 2018) and in the North Pacific

Ocean (144–574 $cm^{-3}$, Schulze et al., 2020). Note that $d_{act}$ and $\kappa$ were calculated under the assumption that larger particles were initially activated as CCN; the calculations did not consider uncertainties associated with the size dependence of the chemical composition and the external mixing state condition.

In previous studies, yearly cluster analysis (Silvergren et al., 2014; Jung et al., 2018; Lange et al., 2018, 2019) showed that the averaged number–size distribution during summer and early autumn in the Arctic was characterized

by a bimodal distribution and a large contribution of biogenic OM from local emissions and aged aerosols. The mean values of $\kappa$ in the summer and early autumn observations conducted in the Arctic Ocean were 0.33 (Martin et al., 2011, over the Arctic Ocean), 0.33 (Lange et al., 2018, in Greenland), and $\sim$0.6 (Silvergren et al., 2014, at the Zeppelin station). Our results ($\kappa$: 0.17–0.68) are consistent with previous observations in the marine boundary layer (0.1–0.96, Hendrickson et al., 2021), in the marine environment including the North Pacific Ocean (0.27–

0.72, Schulze et al., 2020; 0.49–0.86, Kawana et al., 2022b), and the modeled global $\kappa$ in the marine environment ($0.72 \pm 0.24$, Pringle et al., 2010). A wide range of observed CCN AF at around 0.4 % SS has been reported (40–80 %, Silvergren et al., 2014; 40–60 %, Jung et al., 2018; 71 %, Lange et al., 2018) depending on the emission source, with a tendency for the value to be lower when the fraction of fine particles and biogenic OM was large. In terms of the autumn season and the corresponding hygroscopicity parameters determined in this study, the organics

likely contributed to CCN activation during P2 and P4 via biological activity, while the contributions of sea salt and sulfate were dominant over the Arctic Ocean during P3. These results are consistent with previously obtained





observations in the Arctic, i.e., primary and secondary OM with less hygroscopicity, associated with ice melting and biological activity, contributed to CCN activation during the summer, while CCN activation was mainly determined by highly hygroscopic components during winter, when the contribution of sea salt was large and biological activity was limited owing to the low temperature (Kawana et al., 2022a). During the spring pre-bloom

periods in the North Pacific Ocean, CCN activity was characterized by less hygroscopic OM (estimated $\kappa$: 0.13–0.29) proportional to the increase in Chl-*a*, especially in terms of the fine particles with a relatively high fraction (37 %) of biogenic OM in $PM_1$ (Kawana et al., 2022b). It should be considered that size-resolved chemical composition analysis conducted in previous studies indicated that OM and sulfate are enriched in fine particles (<1 µm), while sea salt is enriched in coarse particles (i.e., Rinaldi et al., 2009). Chemical analysis of maritime Arctic

aerosol particles in the non-refractory components in submicron particles using an aerosol mass spectrometer showed that the average mass fraction of OM and sulfate accounted for 20–22 % and 21–35 %, respectively, and that estimated for sea salt was 37–47 % in the Polar regions (Arctic and Antarctic, Schmale et al., 2013; Ovadnevaite et al., 2014). The chemical composition obtained in this study was based on $PM_{2.5}$ particles only, and marine biogenic OM was also present in addition to sea salt and sulfate, potentially with OM and sulfate

representing the major components in the Aitken mode and sea salt representing a major component in the accumulation mode, as inferred from the findings of previous studies. Note that in considering the general enrichment of organics in fine particles around the CCN activation diameter, the fraction of OM might be higher for characterizing the CCN property, especially for fine particles.

The number concentration of INPs (Fig. 8d) exhibited a trend similar to that of both CN and CCN; specifically,

high number concentrations were observed during P1, P4, and P5 (Fig. 7e and 7f). The $N_{INP}$ values activated at temperatures higher than −18, −24, and −30 °C ($N_{INP, -18°C}$, $N_{INP, -24°C}$, and $N_{INP, -30°C}$) were in the range of 0.01–0.09, 0.7–11, and 7–29 $L^{-1}$, respectively, during P1 and P5 with terrestrial influence, and in the range of 0.01–0.09, 0.1–2, and 2–31 $L^{-1}$, respectively, during P2 and P4 with marine influence. The $N_{INP}$ values in the Arctic Ocean (during P3) were in the range of 0.001–0.016, 0.012–0.27, and 0.46–6 $L^{-1}$, implying that pristine marine $N_{INP}$ values are

approximately one order of magnitude lower than elsewhere (i.e., North Pacific Ocean). The $N_{INP}$ values at temperatures higher than −18 and −24 °C (from $10^{-3}$ to $10^{-1}$ $L^{-1}$) in this study were broadly within the range of the measured monthly values over the Arctic Ocean during the MOSAiC campaign (from $10^{-4}$ to $10^{-2}$ $L^{-1}$, from September–November, Creamean et al., 2022), and at the research stations at Alert, Barrow, Ny-Ålesund, and Villum in the Arctic region (from $10^{-2}$ to $10^{-1}$ $L^{-1}$, Wex et al., 2019). Previous studies have reported that $N_{INP}$ in

marine air masses is one–three orders of magnitude lower than $N_{INP}$ in air masses with terrestrial-origin OM and dust that are sources of INPs, while marine biogenic OM can be an important source of INPs in a clean marine



environment, especially at high temperature conditions (above −25 °C) in terms of ice formation (Wilson et al., 2015; DeMott et al., 2016; McCluskey et al., 2017; Welti et al., 2020), or at even higher temperatures of approximately −10 °C (Hartmann et al., 2021, European Arctic) or from −15 to −22 °C (McCluskey et al., 2018b, Mace Head, Atlantic). Conversely, between −25 and −30 °C owing to the activation of mineral dust in this

temperature range, $N_{INP}$ is increased substantially (by approximately two orders of magnitude) based on $N_{INP}$ temperature spectra in the Northern Hemisphere (Welti et al., 2020). Our $N_{INP, -30°C}$ (0.46–6 L$^{-1}$) was slightly lower than the reported values (10–100 L$^{-1}$) in the North Polar region (Welti et al., 2020).

### 3.5. Impact on the association of marine biota to the contributions to CCN and INPs

The relationship between $N_{INP}$ and marine biological sources was determined in this study, and therefore the

observed $N_{INP}$ was characterized in terms of the regressions based on bioindicators and fluorescent aerosols (Figs. 8 and 9, Table 2). The high correlations ($R > 0.97$, $n = 6$, from 9–28 October) were obtained using all marine bioindicators (TEPs, CSPs, and Chl-$a$) when the wind effect was considered, where the terrestrial influence was found to be low. The evident tendency was for $N_{INP, -18°C}$; these marine biogenic materials are very likely involved in the process of INP formation at relatively high temperatures. Even in the analysis without the data point with the

highest values (sample obtained on 28 October 2019), which might have affected this correlation analysis (Fig. S5), high correlation coefficients were obtained especially with CSPs ($R > 0.90$) and with TEPs ($R > 0.91$) for $N_{INP, -24°C}$. The correlation coefficient with Chl-$a$ was somewhat lowered but still significant ($R > 0.72$). The equations of $N_{INP, -24°C}$ (as an example) with marine biota and WS are expressed as follows:

$[N_{INP, -24°C}]$ (particles L$^{-1}$) = (0.003 ± 0.0003) · [TEPs] (µg XGeq L$^{-1}$) · WS (m s$^{-1}$) + (−0.73 ± 0.14) ($R$: 0.98),  (5)

$[N_{INP, -24°C}]$ (particles L$^{-1}$) = (0.005 ± 0.0003)· [CSPs] (µg BSAeq L$^{-1}$) · WS (m s$^{-1}$) + (−0.20 ± 0.063) ($R$: 0.99),  (6)

$[N_{INP, -24°C}]$ (particles L$^{-1}$) = (0.23± 0.022)· [Chl-$a$] (mg m$^{-3}$) · WS (m s$^{-1}$) + (−0.45 ± 0.14) ($R$: 0.98).   (7)

In the same region in November 2018, from a cruise track similar to that of R/V *Mirai* made 1 year earlier, Inoue et al. (2021) reported that INPs were detected at temperatures higher than −14 °C in the Arctic at low altitudes

when the aerosol particles were characterized by a high mass fraction of OC. Their meteorological analysis implied that marine-derived OM with sea salt was supplied to the atmosphere by high waves and strong winds. Several tank experiments and in situ observations using seawater samples obtained during periods of high biological activity suggested that high concentrations of TEPs and bacteria in the sea surface migrate and transfer to atmospheric aerosols and cloud water, and become a major constituent of biogenic INPs at temperatures above −25 °C (van

Pinxteren et al., 2020, 2022; Santander et al. 2021, 2022; Mitts et al., 2021). Notably, the number of data pairs (*n*





= 5 or 6) was limited in this study because the observations of INPs and marine biological indicators did not

coincide in some cases. Clearly, future verification with a larger number of observational points is recommended.

Figure 9 and Table 3 show that both $N_{CCN}$ and $N_{INP}$ correlated more strongly for fine fluorescent particles ($1 < D_p <$

2.5 μm) ($R$: 0.50–0.71 for marine air masses) than for coarse fluorescent particles ($R$: 0.19–0.26, $D_p > 2.5$ μm). This

tendency is similar to the relationship between fluorescent particles and bioindicators (see Sect. 3.3). Finally, we

propose equations to estimate $N_{INP, -T}$ with all fluorescent particles (FL$_{all}$), which can be expressed as follows:

$[N_{INP, -18°C}]$ (particles L$^{-1}$) = (0.0010) · [FL$_{all}$] $^{0.905}$ (L$^{-1}$) ($R$: 0.50),   (8)

$[N_{INP, -24°C}]$ (particles L$^{-1}$) = (0.0025) · [FL$_{all}$] $^{1.457}$ (L$^{-1}$) ($R$: 0.60),   (9)

$[N_{INP, -30°C}]$ (particles L$^{-1}$) = (0.0467) · [FL$_{all}$] $^{1.553}$ (L$^{-1}$) ($R$: 0.67).   (10)

10        In terms of fluorescence pattern, Type AB and Type ABC showed stronger correlation ($R$: 0.68–0.99), although

their number fraction (10 % and 5 %, respectively) was lower than that of Type A and Type B (45 % and 24 %,

respectively). The absolute density of Type AB or Type ABC was still more abundant than that of INPs, suggesting

that only a certain proportion of fluorescent bioaerosols would activate as INPs. Following a study in Vancouver

Island, Mason et al. (2015) also reported that particles that become active as INPs at a temperature of between −15

and −25 °C have a large contribution from biological particles, based on measurements of fluorescent particles with

a WIBS-4A with high linear correlation ($R = 0.83$) between $N_{INP, -25°C}$ and FL$_{all}$ over a wide size range ($D_p$: 0.5–10 μm).

This might be in accord with the results mentioned above that suggested that a large contribution of biological particles

become active as INPs at high temperatures (from −10 to −25 °C) with marine biogenic sources in the remote ocean

(Wilson et al., 2015; McCluskey et al., 2017, 2018b; Welti et al., 2020). Note that our analysis did not include

particles smaller than 1 μm owing to uncertainty of the fluorescence measurements, and the number concentration

of fluorescent particles was an order of magnitude lower than that of Mason et al. (2015) in the middle latitudes.

Generally, CCN are present mainly in fine particles such as PM$_1$, except for giant CCNs, and thus are more strongly

affected by OM, which is more abundant in fine particles (Christiansen et al., 2020). In contrast, particles that can

act as INPs are generally dominated by coarse particles with lower hygroscopicity, such as dust, mineral particles,

and volcanic ash (Murray et al., 2012), which might be less affected by the presence of OM in the fine particles.

Consequently, CCN and INPs might consist of particles of different size and chemical composition that become

activated, with biogenic particles including both CCN and INPs with moderate hygroscopicity involved in the

activation processes. For example, during the mesocosm experiment, with high concentrations of Chl-$a$, bacteria,

and phytoplankton, Prather et al. (2013) reported that OM (i.e., organic gel particles) with sea salt were dominant

in the fine particles (diameter: <1 μm), whereas sea salt and biological particles (i.e., bacteria) were dominant in

the coarse particles (diameter: >1 μm). Moreover, activation of CCN and INPs was suppressed in high



concentrations of total organic carbon with reduction in hygroscopicity, possibly owing to an organic film that inhibited the release of particles as sources as INPs (Prather et al., 2013). The CCN-derived $\kappa$ values of particles in the Aitken and accumulation modes were 0.21–0.29 and 0.57, respectively, indicative of a large contribution of biogenic OM and sulfate in submicron particles, whereas sea salt and to a lesser extent dust were the main

contributors in super-micron particles. Moreover, the values of $N_{\mathrm{INP}}$ activated at higher temperatures (−15 °C) suggested derivation from local biogenic sources (Gong et al., 2019). Cloud chamber studies with variable conditions of water SS and temperature also showed that INP activation (not less CCN particles) might be suppressed by competition for water vapor because the more active CCN particles initially take up all the water (Simpson et al., 2018). These studies revealed that the temporal evolutions of CCN and INPs are complex in the

natural environment and depend on their individual properties (i.e., chemical composition, mixing state, particle size, and number concentration). Recently, climate models representing biogenic sources and cloud formation on the global scale have evolved to incorporate a series of processes that include the abundance and transport of biological material that can be sources of CCN and INPs over the land and sea (Burrows et al., 2009a, 2009b), adsorption of different types of OM (i.e., polysaccharides, proteins, and lipids) in the sea surface layer and SSA

formation (Burrows et al., 2014, 2016), and the contribution to INP activation by biological particles in comparison with that of dust (Burrows et al., 2013, 2022). Models estimate that the abundance of bioaerosols is lower in the marine environment than over land, but they remain a major source of INPs in locations unaffected by dust or after long-distance transport of air masses from land in the high-latitude polar regions, which is consistent with observations of INP number concentrations in terrestrial and marine areas (McCluskey et al., 2018c; Welti et al.,

20   2020).

Our results showed for the first time that all of the bioindicators, fluorescent particles, INPs, and CCN correlated positively, indicating that marine biological materials contributed substantially as a source of bioaerosols and cloud particles, over the remote Bering Sea and the Arctic Ocean during the autumn bloom. This suggests the possibility that these biological materials were transferred to marine atmospheric aerosols, and were directly involved as CCN

and INPs in the maritime air masses, although some contribution from terrestrial sources might have existed. $N_{\mathrm{CCN}}$ and $N_{\mathrm{INP}}$ showed strong correlation ($R$: 0.77–0.86), and the correlations with the biological indicator (and wind) and fluorescent particles were similar (Tables 2 and 3), suggesting that marine biological sources and the formed fluorescent marine bioaerosols might contribute to cloud formation via both CCN and INPs when biological sources are dominant. To constrain the effects of partitioning between CCN and INPs on number concentration and

activation, including other factors such as the selective and size-dependent transfer of biogenic OM from the sea



surface to the aerosol phase and regulation of kinetics in the processes, additional observational data and experimental verification will be required.

## 4. Conclusions

We evaluated potential links of sea surface biological activity (considering TEPs, CSPs, and Chl-*a* as indicators) to atmospheric fluorescent bioaerosols, CCN, and INPs based on atmospheric and seawater sampling and online observations conducted in the North Pacific Ocean, the Bering Sea, and the Arctic Ocean during September–November 2019. According to the air mass origins and the associated aerosol chemical compositions, the observations were categorized into five periods. During P1 and P5 over the North Pacific Ocean, largely influenced

by long-range transport from terrestrial regions including the Asian continent, particle chemical compositions were characterized by high mass fractions of Organics and sulfate (15–22 % and 28–48 %, respectively). In contrast, during P2, P3, and P4 over the Bering Sea and the Arctic Ocean, mainly influenced by maritime air masses, high mass fractions of sea salt (43–88 %) were noted. The concentrations of nutrients and bioindicators (TEPs, CSPs, and Chl-*a*) in the seawater from the North Pacific Ocean and the Bering Sea (up to 1.3 mg m$^{-3}$ of Chl-*a*) were high,

suggesting high biological activity (e.g., an autumn bloom) and associated locally produced biogenic OM. The average number concentration of fluorescent bioaerosols in the range of $1 < D_p < 2.5$ μm and $D_p > 2.5$ μm was 25 and 7 particles L$^{-1}$, respectively, with predominance of Types A, B, AB, and ABC in the fluorescence pattern.

    The averaged aerosol number concentration over the North Pacific Ocean, the Bering Sea, and the Arctic Ocean was 231–356, 63–184, and 80 cm$^{-3}$, respectively, and the corresponding averaged $N_{CCN}$ at 0.4 % SS was 99–151,

43–139, and 36 cm$^{-3}$, respectively. Over the North Pacific Ocean (P1 and P5), higher aerosol number concentrations were observed, with particular dominance of fine particles in the Aitken mode. Over the Bering Sea and Arctic Ocean (P2, P3, and P4), lower aerosol number concentrations were observed with a major peak in the accumulation mode. The CCN activation parameters in each period were characterized by the air mass origin and chemical composition. During P1, the CCN-derived $\kappa$ value was as low as 0.17 and the CCN activated fraction at 0.4 % SS

was approximately 50 %, with a high mass fraction of OM and sulfate, suggesting a large contribution of less hygroscopic components from terrestrial/anthropogenic sources. During P2, when local marine sources were dominant, the $\kappa$ values were also as low as 0.42, and primary gel organic particles with less solubility (e.g., TEPs and CSPs) were suggested to contribute substantially to the reduction in particle hygroscopicity. Conversely, during P3 and P4, the CCN activation parameters were characterized by high values of $\kappa$ (0.67–0.68) and high CCN AFs



(64–76 %), indicating that hygroscopicity and CCN activity were controlled by the highly hygroscopic components from natural sources such as sea salt and sulfate in addition to biogenic OM. The $N_{INP}$ measured at −18 and −24 °C during P2 and P4 (with influence from marine sources of SSAs, including biogenic OM) were 0.01–0.09 and 0.1–2 L$^{-1}$, respectively, while those over the Arctic Ocean (during P3) were 0.001–0.016 and 0.012–0.2 L$^{-1}$, respectively.

The $N_{INP}$ values measured at −18 and −24 °C (from $10^{-3}$ to $10^{-1}$ L$^{-1}$) over the Arctic Ocean were broadly within the range of reported values for the Arctic with predominance of marine biogenic emission influences.

During periods with predominant marine biological activity, we assessed the impact of marine biota on the formation of fluorescent bioaerosols and CCN/INPs by examining the correlations among them. We found that the number density of bioaerosols correlated strongly with all types of oceanic bioindicator, i.e., TEPs, CSPs, and Chl-

*a* (*R*: 0.81–0.88) when considering the wind-uplifting effect from sea surface to the atmosphere, which suggests a strong link between marine biota and marine bioaerosols. Furthermore, the $N_{INP}$ values measured at relatively high temperatures (−18 and −24 °C) correlated positively with the three oceanic bioindicators when local WS was considered (*R*: 0.58–0.95 and 0.79–0.93, respectively), even when the extreme outlier point was omitted, and they also correlated positively with total fluorescent bioaerosols (*R*: 0.50 and 0.60, respectively), particularly with those

in Type AB and Type ABC (*R*: 0.90–0.99 and 0.68–0.76, respectively). Our results demonstrate that bioindicators, fluorescent particles, INP, and CCN correlate positively in any combination, indicating that marine biota contribute substantially as a source of bioaerosols and cloud formation via INP and CCN over the remote Arctic Ocean during periods of high biological activity.

**Acknowledgments**

We acknowledge the assistance from the captain and crew of R/V *Mirai* and the support from Marine Works Japan, Ltd. and Nippon Marine Enterprise, Ltd. This research was supported by the Ministry of Education, Culture, Sports, Science, and Technology (MEXT) and MEXT/JSPS KAKENHI (grant number No. JP18H04143 and 21H04933) and the Arctic Challenge for Sustainability II (ArCS II), Program Grant Number JPMXD1420318865. We thank

James Buxton MSc, from Edanz (https://jp.edanz.com/ac), for editing a draft of this manuscript.

**Data availability**

The data discussed in this manuscript are available from the following websites: JAMSTEC (2019) MIRAI MR19-03C Cruise Report and Data Book and https://www.godac.jamstec.go.jp/darwin_tmp/explain/81/j/.





**Author contributions**

KK and YK designed the research, and KK wrote the manuscript and prepared all figures. FT performed the cruise observations and data collection including sampling with contributions from KM, TM, YK, KK, YT, and YI. KK

performed the analysis of fluorescent particles and CCN activity in the atmosphere with contribution from FT and YI, respectively, and bioindicators in the seawater with contribution from KM. TM analyzed and provided the data of number–size distribution and number concentrations. YT analyzed and provided the data on the INP properties. YK analyzed and provided the data on the O₃ and CO concentrations. AI performed the model calculation to identify the sources.

**Competing interests**

The authors declare that they have no conflict of interest.

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





**Figures/Tables**

**Table 1.** Summary of observation parameters in each period. *FL (fine): fluorescent particles with diameter of >1 µm (1 < $D_p$ < 2.5 µm), FL (coarse): fluorescent particles diameter of >2.5 µm ($D_p$ > 2.5 µm).

| | | P1 09/30-10/06 | P2 10/07-10/09 | P3 10/10-10/27 | P4 10/28-11/02 | P5 11/03-11/07 |
|---|---|---|---|---|---|---|
| *Aerosol* | | | | | | |
| Mass fraction | OC | 22% | 4% | 5% | 15% | 15% |
| (PM$_{2.5}$) | Sulfate | 48% | 5% | 14% | 31% | 28% |
| | Seasalt | 19% | 88% | 76% | 43% | 46% |
| FL (fine) | L$^{-1}$ | 22 ± 12 | 24 ± 13 | 19 ± 10 | 23 ± 9 | 36 ± 20 |
| FL (coarse) | L$^{-1}$ | 9 ± 5 | 5 ± 2 | 5 ± 3 | 7 ± 4 | 10 ± 6 |
| $N_{total}$ | cm$^{-3}$ | 231 ± 143 | 63 ± 30 | 80 ± 50 | 184 ± 72 | 356 ± 238 |
| $N_{CCN, 0.4\%SS}$ | cm$^{-3}$ | 99 ± 24 | 43 ± 22 | 36 ± 13 | 139 ± 44 | 151 ± 85 |
| $K_{CCNC, 0.4\%SS}$ | | 0.17 ± 0.06 | 0.42 ± 0.04 | 0.67 ± 0.21 | 0.68 ± 0.02 | 0.60 ± 0.06 |
| CCN_AF | | 0.66 ± 0.19 | 0.63± 0.16 | 0.64 ± 0.16 | 0.76 ± 0.17 | 0.48 ± 0.11 |
| $N_{INP, -30C}$ | L$^{-1}$ | 18 ± 16 | 3 ± 2 | 5 ± 3 | 22 ± 13 | 7 |
| *Seawater* | | | | | | |
| TEP | µg XGeq L$^{-1}$ | 62 ± 7 | 77 ± 26 | 45 ± 9 | 79 ± 4 | 61 ± 1 |
| CSP | µg BSAeq L$^{-1}$ | 39 ± 3 | 21 ± 3 | 6 ± 4 | 35 ± 6 | 16 ± 2 |
| Chl-$a$ | mg m$^{-3}$ | 0.53 | 1.08 ± 0.32 | 0.3 ± 0.1 | 0.84 ± 0.33 | 0.77 |





**Table 2.** Correlation coefficients ($R$) between the number concentrations of fluorescent bioaerosol particles, CCN, and INPs in the atmosphere and WS and/or bioindicator concentrations. *FL (fine): fluorescent particles with diameter of >1 µm ($1 < D_p < 2.5$ µm), FL (coarse): fluorescent particles with diameter of >2.5 µm ($D_p > 2.5$ µm).

5 Note that in the correlation analysis of INPs and marine biological indicators, the number of data pairs was limited ($n = 6$). Refer to Fig. S4 for the results of additional analysis excluding the data point with the highest values (sample obtained on 28 October 2019), which might have affected the correlation.

| | | FL (fine) | FL (coarse) | $N_{CCN}$ | $N_{INP,-18C}$ | $N_{INP,-24C}$ | $N_{INP,-30C}$ |
|---|---|---|---|---|---|---|---|
| All (P1-P5) | WS | 0.54 | 0.51 | | | | |
| | TEP | 0.02 | 0.06 | 0.76 | 0.75 | 0.83 | 0.81 |
| | CSP | 0.06 | 0.05 | 0.76 | 0.62 | 0.86 | 0.82 |
| | Chl-a | 0.37 | 0.3 | 0.70 | 0.82 | 0.75 | 0.74 |
| | TEP × WS | 0.85 | 0.84 | 0.72 | 0.82 | 0.96 | 0.99 |
| | CSP × WS | 0.68 | 0.59 | 0.76 | 0.75 | 0.97 | 0.96 |
| | Chl-a × WS | 0.85 | 0.71 | 0.75 | 0.90 | 0.96 | 0.98 |
| | TEP × WS² | 0.86 | 0.77 | 0.63 | 0.78 | 0.93 | 0.98 |
| | CSP × WS² | 0.84 | 0.65 | 0.71 | 0.78 | 0.97 | 0.98 |
| | Chl-a × WS² | 0.86 | 0.67 | 0.68 | 0.84 | 0.95 | 0.99 |
| Marine/Arctic-Bering(P2-P4) | | | | | | | |
| | WS | 0.58 | 0.53 | | | | |
| | TEP | 0.01 | 0.03 | 0.83 | 0.85 | 0.83 | 0.83 |
| | CSP | 0.08 | 0.04 | 0.91 | 0.97 | 0.94 | 0.95 |
| | Chl-a | 0.33 | 0.24 | 0.72 | 0.83 | 0.77 | 0.77 |
| | TEP × WS | 0.88 | 0.82 | 0.95 | 0.98 | 0.98 | 0.99 |
| | CSP × WS | 0.81 | 0.62 | 0.96 | 0.99 | 0.99 | 0.99 |
| | Chl-a × WS | 0.85 | 0.66 | 0.94 | 0.99 | 0.98 | 0.98 |
| | TEP × WS² | 0.90 | 0.80 | 0.94 | 0.97 | 0.98 | 0.99 |
| | CSP × WS² | 0.93 | 0.71 | 0.97 | 0.99 | 0.99 | 0.99 |
| | Chl-a × WS² | 0.88 | 0.69 | 0.96 | 0.99 | 0.99 | 0.99 |





**Table 3.** Correlation coefficients ($R$) between number concentrations of CCN and INPs and fluorescent bioaerosol particles with different fluorescence patterns. *FL (fine): fluorescent particles with diameter of >1 μm (1 < $D_p$ < 2.5 μm), FL (coarse): fluorescent particles with diameter of >2.5 μm ($D_p$ > 2.5 μm).

| | | $N_{CCN}$ | $N_{INP,-18C}$ | $N_{INP,-24C}$ | $N_{INP,-30C}$ |
|---|---|---|---|---|---|
| All (P1-P5) | FL (fine) | 0.62 | 0.60 | 0.50 | 0.54 |
| | FL (coarse) | 0.47 | 0.28 | 0.12 | 0.16 |
| * FL (fine) | Type A | | 0.37 | 0.11 | 0.11 |
| | Type B | | 0.39 | 0.06 | 0.06 |
| | Type C | | 0.01 | 0.003 | 0.02 |
| | Type AB | | 0.71 | 0.92 | 0.98 |
| | Type BC | | 0.001 | 0.005 | 0.001 |
| | Type ABC | | 0.37 | 0.67 | 0.76 |
| | $N_{CCN}$ | | 0.86 | 0.85 | 0.77 |
| Marine/Arctic -Bering (P2-P4) | FL (fine) | 0.63 | 0.50 | 0.60 | 0.67 |
| | FL (coarse) | 0.35 | 0.17 | 0.15 | 0.21 |
| * FL (fine) | Type A | | 0.15 | 0.17 | 0.21 |
| | Type B | | 0.26 | 0.32 | 0.40 |
| | Type C | | 0.01 | 0.02 | 0.01 |
| | Type AB | | 0.90 | 0.99 | 0.99 |
| | Type BC | | 0.001 | 0.001 | 0.005 |
| | Type ABC | | 0.68 | 0.76 | 0.80 |
| | $N_{CCN}$ | | 0.96 | 0.98 | 0.97 |

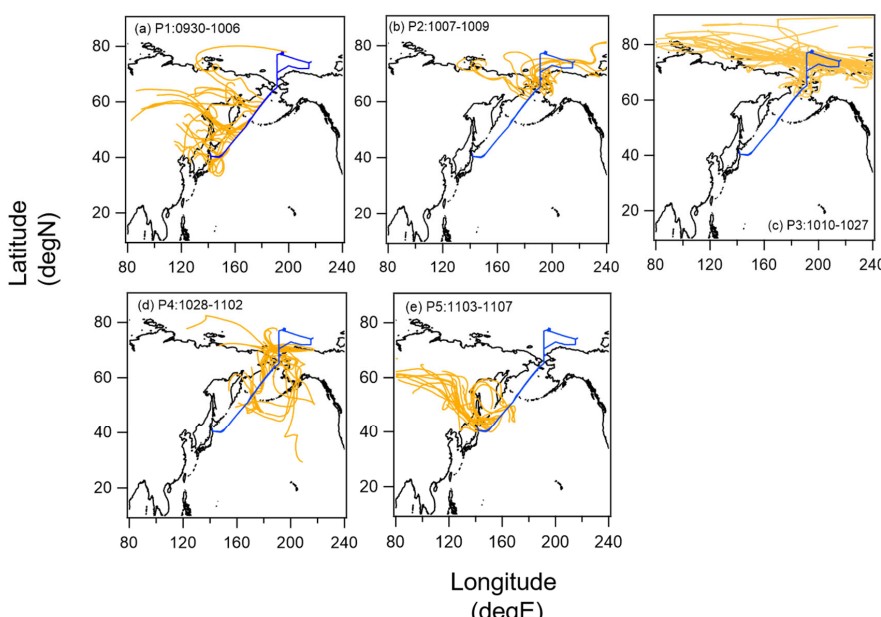

**Figure 1.** Five-day backward trajectories of air parcels along with the cruise track (0000, 0600, 1200, and 1800 UTC) for (a) Period 1 (P1; 30 September to 6 October 2019), (b) Period 2 (P2; 7–9 October 2019), (c) Period 3 (P3; 10–27 October 2019), (d) Period 4 (P4; 28 October to 2 November 2019), and (e) Period 5 (P5; 3–7 November 2019). Blue line indicates the cruise track.

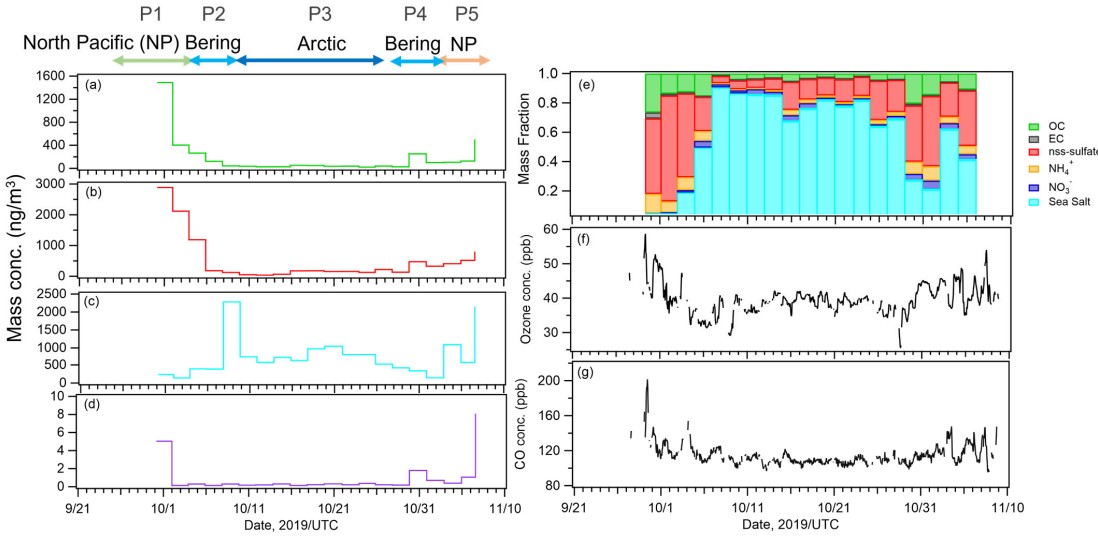

**Figure 2.** Time series of mass concentrations of (a) OC, (b) non-sea-salt sulfate (nss-sulfate), (c) sea salt, and (d)
5  levoglucosan. Time series of (e) mass fractions of chemical components: OC, EC, nss-sulfate, ammonium, nitrate,
and sea salt. (f) O$_3$ and (g) CO concentrations during the observation period.



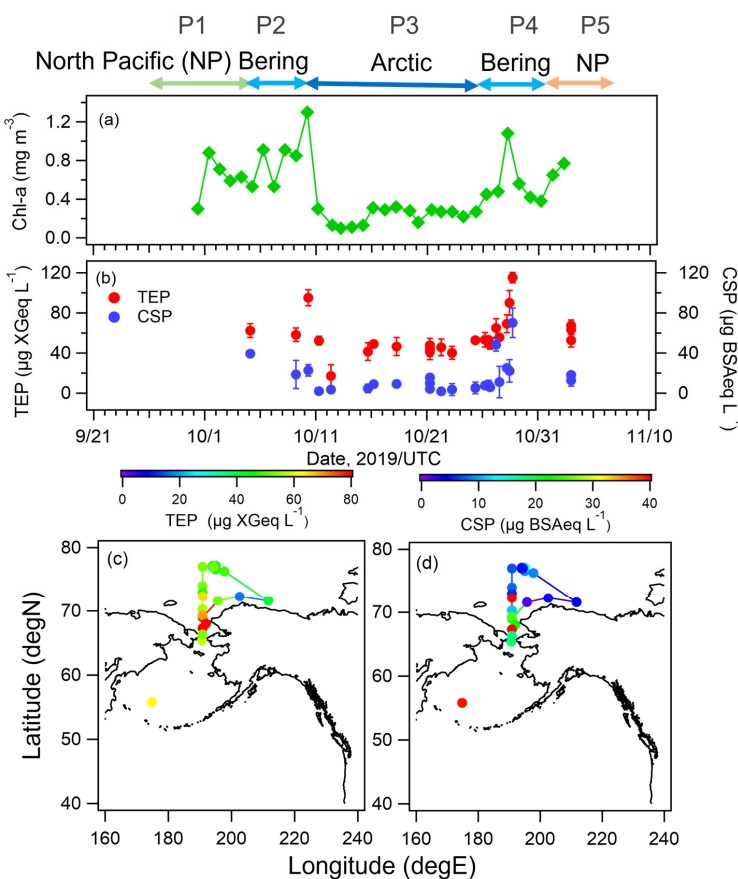

**Figure 3.** Time series of concentrations of (a) Chl-*a*, (b) TEPs, and CSPs, and geographic distributions of (c) TEPs, and (d) CSPs during the observation period. Error bars represent one standard deviation.

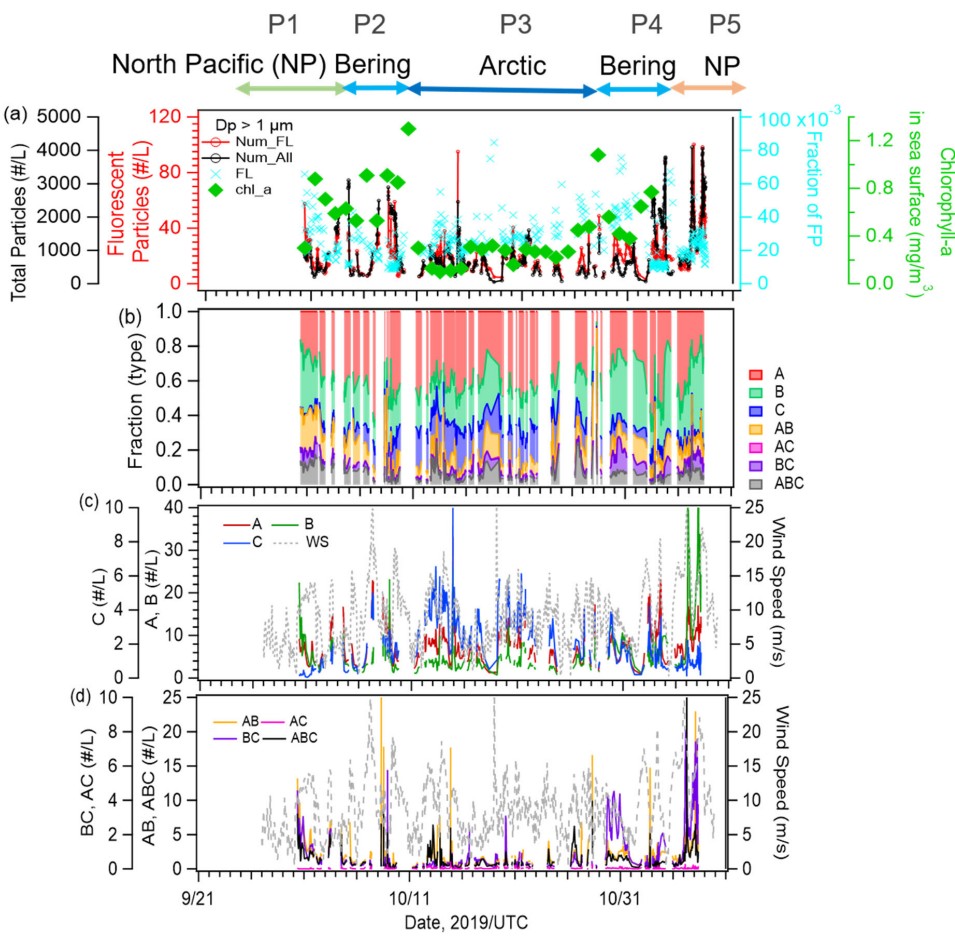

**Figure 4.** Time series of (a) number concentrations of total particles (black line) and fluorescent aerosol particles (red line), and number fractions of FAPs (blue marker), (b) relative fractions of particle types with a fluorescence pattern, (c) number concentrations of Type A, B, and C (second axis) particles, and (d) number concentrations of Types AB, BC, AC, and ABC. Dashed lines in (c) and (d) represent local WS.





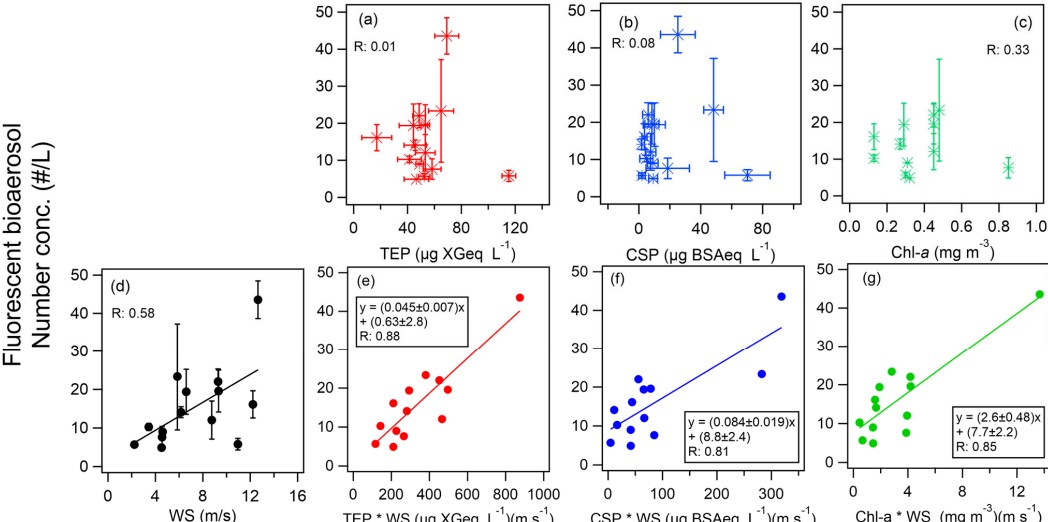

**Figure 5.** Scatter plots of fluorescent bioaerosols and bioindicators as (a) TEPs, (b) CSPs, and (c) Chl-*a*, and scatter plots of fluorescent bioaerosols and (d) WS, and equations of number concentrations of fluorescent bioaerosols as function of the product of WS and the bioindicators of (e) TEPs, (f) CSPs, and (g) Chl-*a*. Colored lines represent the orthogonal regression lines. Error bars represent one standard deviation.





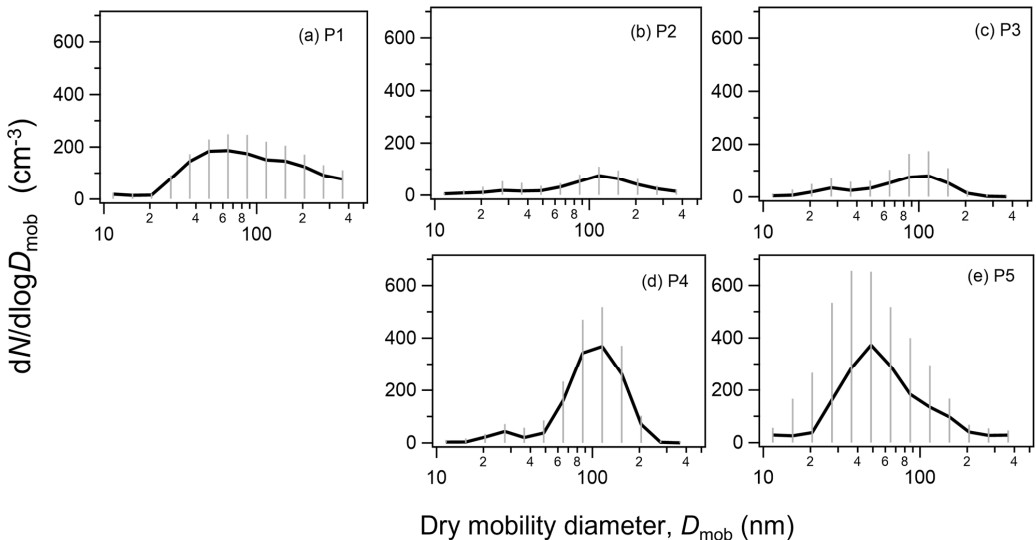

**Figure 6.** (a) Aerosol number–size distributions during (a) P1, (b) P2, (c) P3, (d) P4, and (e) P5. Gray bars represent one standard deviation.

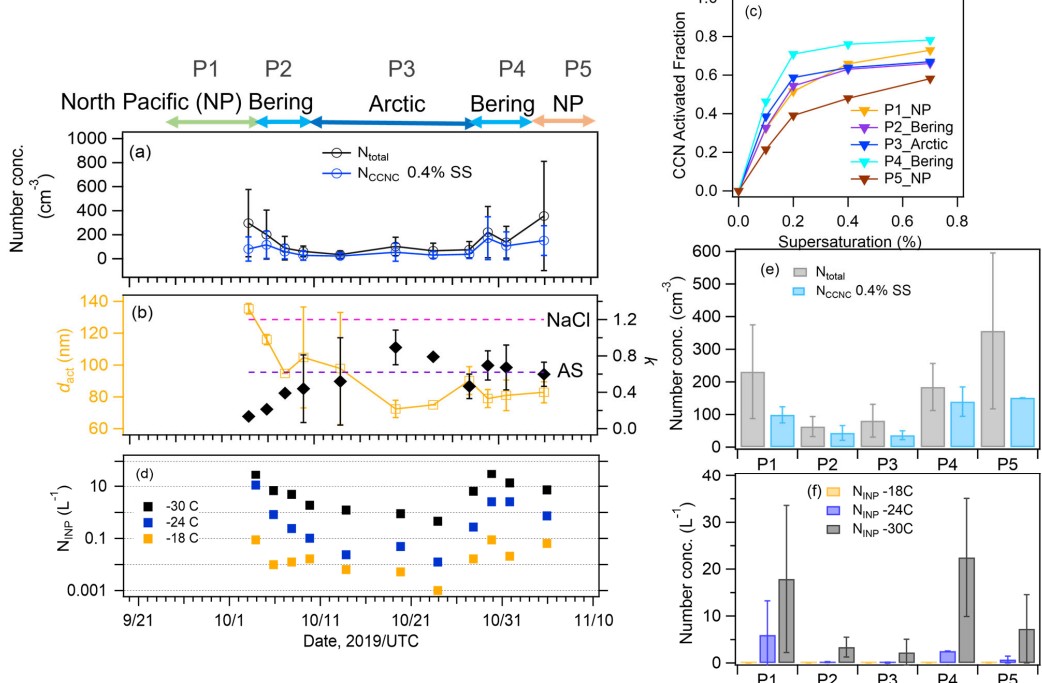

**Figure 7.** Time series of number concentrations of (a) total particles and CCN at 0.4 % SS, and (b) CCN activation diameter ($d_{act}$) and hygroscopic parameter $\kappa$. (c) The AF as a function of SS in each period. (d) Time series of number concentrations of INPs measured at temperatures higher than −18, −24, and −30 °C, and the number concentrations of (e) total particles, CCN, and (f) INPs in each period. Error bars represent one standard deviation.



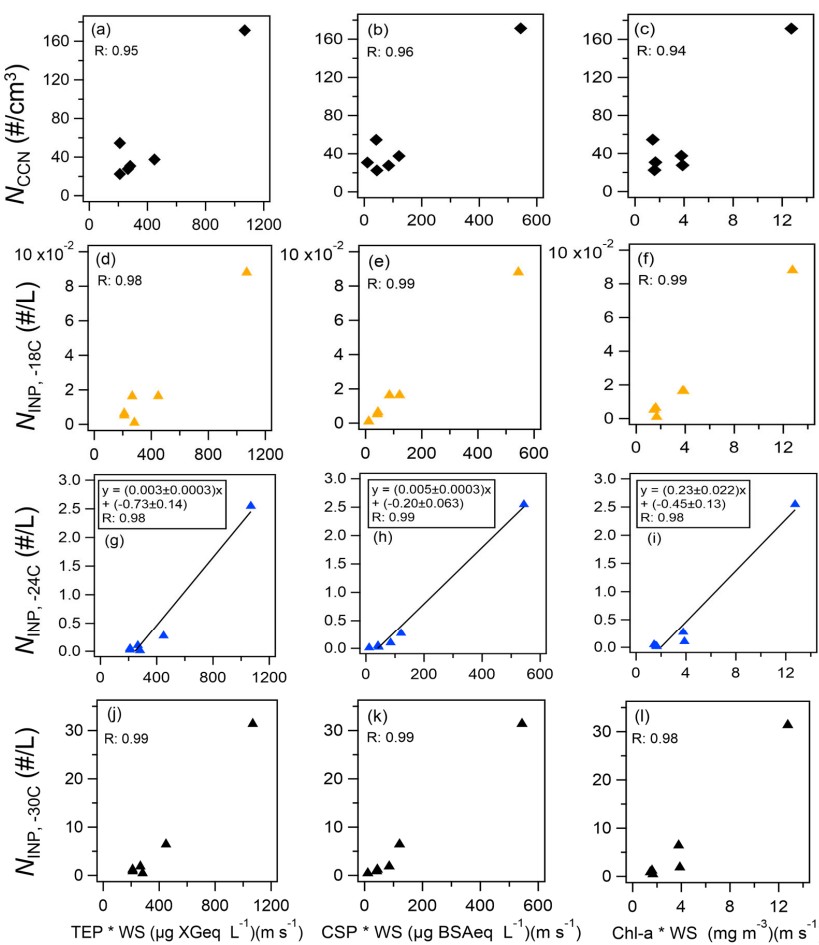

**Figure 8.** Scatterplots of bioindicator concentrations (TEPs, CSPs, and Chl-*a*) multiplied by WS versus (a)–(c) CCN, and INP number concentrations measured at temperatures higher than (d)–(f) −18 °C, (g)–(i) −24 °C, and (j)–(l) −30 °C. Black lines for active INP particles at −24 °C represent the orthogonal regression lines as examples.





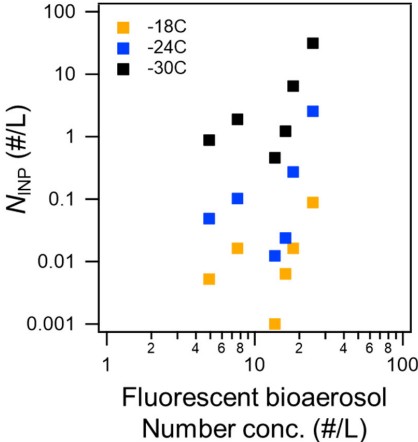

**Figure 9.** Scatterplot of fluorescent bioaerosols (total fluorescent particles) and INP number concentrations measured at temperatures higher than −18, −24, and −30 °C.