# Peer review of "Roles of marine biota in the formation of atmospheric bioaerosols, cloud condensation nuclei, and ice-nucleating particles over the North Pacific Ocean, Bering Sea, and Arctic Ocean"

_Atmospheric Chemistry and Physics, 2023_

## Referee Comment (RC1)

The article aims to relate certain marine bioindicators with aerosolised fluorescent particles and the presence of active aerosols such as CCN and INPs.

It is interesting the analysis carried out in the different localities and how they are affected by different air masses that have a strong influence on the results obtained for bioaerosols.

The article can be published if the following major and minor changes are made:

A. Major changes:
- Specify in the introduction, cruise observations and at the beginning of the corresponding section of the results, which bioindicators have been used. In the case of Chl-a, also add why was used it.

- The suggestion of new equations for the different calculations is good, but the components of each equation should be better explained, for example in the pag. 10.

- In "Cruise Observations section 2.2":

    ▪ it is not explained how the filters used for chemical or INP analysis are processed. Only for the INP analysis, a vague reference is made to two articles, but I suggest explaining briefly in the text how this was done in this study. So, please explain the processing of both kind of filters: quartz and polycarbonate membrane filters.

    ▪ I also recommend to specify where the TEPs and CSPs analyses were done: on the ship or in the laboratory after the campaign? If they were conducted in the laboratory after the campaign, how the samples were conserved until their processing? Moreover, in the line 30 when it's specified "after 1 min", what do you mean: after 1 min of each rinsed time or what specifically?

    ▪ Also, how much seawater was used for Chl-a and nutrient analyses?

- Figure 2: please also include the representation of the different periods P1-P5 at the top of the right section (figs. 2e, 2f, 2g) to better see the intervals of the values.
  And add in the caption of the figure, the abbreviations used in the graphs, so that reader doesn't need to look up the meaning every time in the text.

- Figure 3: when this figure is explained in the text (pag. 8, lines: 10-13), it is stated that figs. 3c and 3d also show Chl-a dispersion, but they only show the dispersion of TEP (3c) and CSP (3d). Please modify it.
  Also in the text, it's a bit confusing when the average values are mentioned, since the Chl-a values are included. In general, the structure of the paragraph is not very clear. Could it be possible to specify better the results, mentioning the correct figs and restructuring a bit the order of the values? For example, figs 3c and 3d are commented before than 3a and 3b.

- Pag 10: Line 2: Could you specify a reference where the DNA staining method is conducted? Or did you do this analysis? If yes, could you include it in the results or supplementary section?

- Page 11: The results obtained with CSPs are not discussed. Please add a brief summary.

- Section 3.4: Lines 2 and 3. Please make a revision of the text, since the commented values for the fig. 6 do not appear to be the same as those represented. In addition, the highest peaks are for P4 and P5 and not for P1 as indicated in the text.

- Figure 7: Graphs a) and e) represent the same data. Would it be possible to delete one of them? It is a bit confusing.
  Also, specify the meaning of the abbreviation "AS" in the figure caption.

- Section 3.5: Pag 15: it is sometimes difficult to follow in the text to which graphs the mentioned values refer. I would recommend adding the figure or table number in brackets after each explanation of the values in the text.
  Similar happens for figure 9 in page 16, it's not commented or specified the values. Furthermore, fig 9. shows all fluorescent particles together and does not distinguish between fine and coarse, whereas it can be deducted from the text that fig 9. shows them separately. Please, explain it better.

- A list of abbreviations is necessary.

B. Minor changes:
  - **Abstract:**
  Last line of abstract, change "contributed" by "point", "define".

  - **Introduction:**
  Pag 4:
    - Line3: specify in brackets what bioindicators you go to analyse: TEP, CSP and Chl-a. If it's the first time that Chl-a is mentioned, also write the complete word.

  - **Results and Discussion:**
  Pag 7:
    - Line 26: where are represented the plots of mass concentrations of OM and $Na^+$ mentioned in the text? Specify it in parenthesis.
    - Line 28: could you mention a brief explanation why the OC, sulfate and sea salt were found in the period P4?
  Pag 8:
    - Line 20: What do you mean by "in this area"? Do you mean the P2 and P4 periods? It is not clear.
    - Line: 28: same as in line 20 when mentioning: "in this period".

  Pag 14:

- Line 25: the word "elsewhere" can refer to literally another place. Please specify whether it refers to another place on land, marine...

- **Conclusions:**
  Pag 19: You explain the periods P1-P4, but you forgot to mention the P5. Could you add something about this period?

C. Typographical corrections:

1. **Introduction:**
   Pag.2:
   - Line 11, change "and" by "or"
   - Line 17: full stop after the parenthesis.
   - Line: 19: close parenthesis after *TEP*; When you mention "microlayer" at the end of the line, which one do you mean, "microlayer of water"? Specify it.
   Pag.3:
   - Line: 6: change "biomaterials" by "biological organisms"
   - Line 21: in the Mediterranean Sea there is another study where INPs are analysed. It was carried out on an island in the Mediterranean Sea. Then, if you are referring to analyses carried out on a ship, don't mention it, but if you are referring to other analyses of bioparticles of marine origin where INP was analysed, mention that too. The reference is: Tang, K., Sánchez-Parra, B., Yordanova, P., Wehking, J., Backes, A. T., Pickersgill, D. A., Maier, S., Sciare, J., Pöschl, U., Weber, B., and Fröhlich-Nowoisky, J.: Bioaerosols and atmospheric ice nuclei in a Mediterranean dryland: community changes related to rainfall, Biogeosciences, 19, 71–91, https://doi.org/10.5194/bg-19-71-2022, 2022

   Pag 4:
   - Line 1: add a "s" to word "cycle".
   Pag 10:
   - Line 16: Rewrite it, specifying that the high correlation appears for "most" of bioindicators, or mention that" for TEP is lower than for CSP and Chl-a".
   - Lines 24 and 25: "the correlation coefficients were almost unchanged when an exponent of 2 was used instead of 1 (see Table 2)".
     Do you mean between which values? Be a little more specific.

2. **Cruise Observations:**
   Pag: 4
   - Line: 9: add a "s" to "measurement"
   Pag 5: specify what is $d_p$ and what is referred as "dry" and "wet" in the equation.

   Pag 7:
   - Line 5: delete the brackets after "respectively".

3. **Results and discussion:**
   Pag 9:
   - Line 11: at the end of the line is said that "…representing 2.5%" Is this value correct? Wouldn't you mean 25%?

If it's correct, in the line 20, is the value 7% correct too? Please, revise it.
- Line 15: add after "marine biogenic source" the following: "(according to our bioindicators analysis)".

Pag 10:
- Line 15: add "fluorescent" before the word "particles"
- Line 26: change "marine" by "fluorescent" or add "fluorescent" between "marine" and "bioaerosols".

Pag 11:
- Line 8: delete "(with bacteria, in some cases)". You don't have evidence of this in the present study.
- Line 15: change "formation" by "detection"
- Line 29: change "study" by "studies"

Pag 12:
- Line 15: it seems that there is an extra space between "Ocean" and "during". Delete the space.

Pag 14:
- Line 19: change Fig. "8d" by "7d"
- Line 20: change Fig "7e" by "7d"

Pag 15:
- Line 11: change Table "2" by "3"
- Line 26: after "…and strong winds" make a full stop.

**4.    Conclusions:**

Pag 19, Line 9: delete "i.e" as you only analysed the three mentioned bioindicators (TEP, CSP and Chl-a) and no more.

---

## Referee Comment (RC2)

**Review of "Roles of marine biota in the formation of atmospheric bioaerosols, cloud condensation nuclei, and ice-nucleating particles over the North Pacific Ocean, Bering Sea, and Arctic Ocean" by Kawana et al.**

**General comment**
This study evaluated if the marine biological activity present in the North Pacific Ocean, Bering Sea, and Arctic Ocean can impact biological aerosol particles formation, and the distribution of CCNs and INPs. To answer this, a large set of measurements were performed between September 27 and November 10, 2019 in an oceanographic cruise. The author found that bioindicators, fluorescent particles, INP, and CCN correlate positively, indicating that marine biota contribute substantially as a source of bioaerosols and cloud formation via INP and CCN over the remote Arctic Ocean during periods of high biological activity. The results are of high importance for this remote, valuable, and uncertain region. The measurements were properly performed and the data analyses are correct and consistent. Although the is well-written several parts can be improved. The present manuscript can be accepted for its publication in ACP after the following comments are properly addressed.

**Major comments:**
1. All the correlations indicated along the manuscript need to be defined as statistically significant or not.
2. The discussion of the present results with literature data needs to be improved avoiding redundancy and trying to be concise and clear.
3. Some figures need to be improved.

**Minor comments:**
P1, L23: Why the CCN concentration for the Arctic Ocean is indicated as a single value (36 cm−3) and not as a range?

P1, L26-27: "The averaged INP concentration ($N_{INP}$) measured at temperatures of −18 and −24 °C with marine sources was 0.01–0.09 and 0.1–2 L−1, respectively" Please indicate the region or period you are referring to.

P3, L6: I am not sure that the term "biomaterials" is appropriate. Please reconsider this.

P6, L20: I am not sure that the term "number densities of INPs" is appropriate. Please reconsider this.

P7, L15: should "P3, 11–27 October 2019" be "P3, 10–27 October 2019" as indicated in Table 1?

P8, L20, L26 and a long the text: I think it would be more appropriate to call "bloom" and "autumn bloom" as "phytoplankton bloom".

P11, L2-3: "suggesting the possibility of dependence on phytoplankton communities in the different oceanic regions". What is reported in the Literature?

P12: In line 13 "Previous studies" is mentioned; however, only Park et al. (2020) is cited. Please add more studies.

P13, L6: I think "such as dicarboxylic acid (i.e., oxalic acid)" should be "such as oxalic acid".

Section 3.4. The discussion related to CCNs is very confused and hard to follow. Please improve this part.

P15, L14: "INP formation". Aerosol particles can form but not an INP. An aerosol particle is capable or not to act as INP.

P17, L3: Why $\kappa$ value for the accumulation mode is indicated as a single value (0.57) and not as a range?

P17, L23-29: I have the impression that this paragraph is repetitive.

P18, L20: Why averaged $N$CCN at 0.4 % SS for the Arctic Ocean is indicated as a single value (36 cm−3) and not as a range?

Tables 2 and 3: indicate if the correlations are statistically significant or not.

Figure 1: The can different periods P1 to P5 be distinguished in the cruise track?

Figure 2: Add labels to panel a, b, c, and d. Also, change the color for EC and NO3 as they look very similar.

Figure 4: I am not sure if panels c and d can be more useful with scatter plots?

Figure 7: y-axis in panel f should go in log scale and y-axis in panel (f) should read "INP concentration".

**Technical comments:**
P2, L5: Add a reference after "processes".
P2, L10: Add a reference after "(INPs)".
P2, L19: "(TEP and protein-containing Coomassie stainable particle (CSP)" It seems that a bracket is missing.
P3, L1: Add a reference after "INPs".
P3, L3-4: "Some CCN grow to giant CCN". Is it not possible to have a primary GCCN?
P3, L11: Add a reference after "observations".
P5, L14: Add a reference after "state".
P8, L13: I think "…contents and suggesting…" should be "…contents, suggesting…"
P9, L12: "total" of what?
P9 L20: "fraction" of what?
P11, L9: Add a reference after "ocean".
P11, L29: "study" should be "studies".

P13, L4: Add a reference after "range".

P13, L32 to P14, L4: Please improve the punctuation here. This is a very long paragraph.

P14, L19: "(Fig. 8d)" should be "(Fig. 7d)"

P14, L25: I think "lower than elsewhere (i.e., North Pacific Ocean)." Should be "lower than in the North Pacific Ocean."

P14, L30: Add a reference after "INPs".

P15, L17: I think "but still significant" should be "but still strong" or something similar.

Figure 4: I think "(black line) and fluorescent" should be "(black line), fluorescent"

---

## Author Comment (AC1)

*Reviewer #1:*

*The article aims to relate certain marine bioindicators with aerosolised fluorescent particles and the presence of active aerosols such as CCN and INPs. It is interesting the analysis carried out in the different localities and how they are affected by different air masses that have a strong influence on the results obtained for bioaerosols. The article can be published if the following major and minor changes are made.*

We would like to thank the reviewer for providing valuable comments and comprehensive views on our work. Our responses are given below. All changes are shown in color in the revised manuscript.

*A. Major changes:*

*- Specify in the introduction, cruise observations and at the beginning of the corresponding section of the results, which bioindicators have been used. In the case of Chl-a, also add why was used it.*

For results of our studies, we have added which indicator we are referring to. We use Chl-a because it is generally used as an indicator of biological activity in a wide range of studies, including observational, satellite, and model studies, and it is easy to apply our equations to model predictions and comparison with other previous studies.

we added the sentences:

 "Besides, chlorophyll a (Chl-a) is well known as a biological surrogate for predicting sea spray organic enrichment and commonly used as an input parameter to combine oceanic biology and atmospheric dynamics with WS (Rinaldi et al., 2013)." (Page 2, Lines 22-24).

"Future studies over different oceanic regions and seasons are required to comprehensively assess the association of bioaerosol evolution with Chl-a, which is derived from satellite observations and/or Earth system models as a proxy for phytoplankton biomass (Ito et al., 2023) and useful for input in model calculations as biological activity." (Page 12, Lines 15-18).

*- The suggestion of new equations for the different calculations is good, but the components of each equation should be better explained, for example in the pag. 10.*

We rearranged previous sentences and added some discussion and implications so that the components in the estimated equations from this study are well explained, including comparisons with previous studies:

For Bioaerosols and bioindicators:

[revised manuscript text omitted]

*- In "Cruise Observations section 2.2":*

*· it is not explained how the filters used for chemical or INP analysis are processed. Only for the INP analysis, a vague reference is made to two articles, but I suggest explaining briefly in the text how this was done in this study. So, please explain the processing of both kind of filters: quartz and polycarbonate membrane filters.*

We have added a detailed description of filter analysis for the chemical composition and INPs as follows:

"For the chemical analysis, the quartz filter was baked at 900℃ for the avoidance of the contamination and packed in the glass bottle before the observation (Taketani et al., 2022). Aliquots of each filter sample were used for the ionic chemical composition, EC/OC analysis, and Levoglucosan. The mass concentrations of ionic species ($NH_4^+$, $Na^+$, $K^+$, $Ca^{2+}$, $Mg^{2+}$, $Cl^-$, $NO_3^-$, and $SO_4^{2-}$) were measured by ion chromatography (model: ICS-1000, Dionex Co., CA, USA), and the mass concentrations of sea salt and non-sea-salt sulfate (nss-$SO_4^{2-}$) were calculated from $Na^+$ assuming the standard seawater composition (Warneck, 2000). The mass concentrations of organic carbon (OC) and elemental carbon (EC) in the $PM_{2.5}$ were also obtained using a thermal/optical carbon analyzer (model: DRI 2001, Desert Research Institute, Reno, NV, USA) with the Interagency Monitoring of Protected Visual Environments protocol, and the mass concentrations of water-insoluble organic carbon were derived by subtraction of the measured water-soluble organic carbon from the total OC. Levoglucosan was analyzed using a derivatization gas chromatography mass spectrometer (model: GCMS-QP2010Plus, Shimadzu Co., Kyoto, Japan)." (Page 6, Lines 9-20).

"A polycarbonate filter (φ: 47 mm) was immersed into MilliQ purified water to prepare a suspension of particle. Then using the particle-containing water droplets with a volume of 5 μL, the number concentrations of INPs upon immersion freezing were obtained using the National Institute of Polar Research Cryogenic Refrigerator Applied to Freezing Test (Tobo, 2016). From the detections made between 0 and −30 °C with a 0.5 °C step, the number concentrations of INPs determined at three selected temperatures (−18, −24, and −30 °C) were used for analysis in this study. The detailed extraction and analysis procedures for INP measurements are described elsewhere (Tobo, 2016; Tobo et al., 2020)." (Page 6, Lines 23-29).

*· I also recommend to specify where the TEPs and CSPs analyses were done: on the ship or in the laboratory after the campaign? If they were conducted in the laboratory after the campaign, how the samples were conserved until their processing? Moreover, in the line 30 when it's specified "after 1 min", what do you mean: after 1 min of each rinsed time or what specifically?*

We filtered seawater and extracted for organic matter and stained on board the ship and analyzed them in the laboratory. We added the details of the processing.

"Surface seawater sampling for TEPs and CSPs was performed using a bucket at 22 sampling stations. In analysis of CSPs, 200 mL of seawater was filtered through a Nuclepore™ polycarbonate membrane filter (cut size: 0.4 μm, Cytiva, Tokyo, Japan) and triplicate filters were obtained from each seawater sample on the ship just after sampling. 1 mL Coomassie brilliant blue staining solution was added to the filter, which was then rinsed five times with 1 mL of Milli-Q® water after 1 min of staining. The filters for CSPs were stored at −40 °C in a freezer. For TEPs, water samples were added formalin to a final concentration of 1% (v/v) after sampling and preserved in a refrigerator (4°C). TEPs samples were then filtered in the laboratory in the same manner as CSPs samples. In the laboratory analysis for TEPs, 1 mL of Alcian blue staining solution, adjusted to pH 2.5, was added to the filter and the filter was rinsed three times with 1 mL of Milli-Q® water after 4 s of staining. filter samples for TEPs were soaked for 2 h in 6 mL of 80 % sulfuric acid for extraction and absorbance was measured at the wavelength of 787 nm (Alldredge and Passow, 1993). For CSPs, 1 mL Coomassie brilliant blue staining solution was added to the filter, which was then rinsed five times with 1 mL of Milli-Q® water after 1 min of staining. Filter samples were soaked for 2 h in 4 mL of 3 % sodium dodecyl sulfate in 50 % isopropyl alcohol with ultrasonic extraction to elute the dye, and the absorbance of the solution was measured at the wavelength of 615 nm (Cisternas-Novoa et al., 2015)." (Page 7, Lines 4-18)

*· Also, how much seawater was used for Chl-a and nutrient analyses?*

We added a description of the amount of seawater used for Chl-a and nutrient analysis.

"Seawater samples collected from the surface were filtered (500 mL ) onto 25 mm Whatman GF/F glass-fiber filters and extracted with N, N-dimethylformamide to determine concentrations

of Chl-a using a fluorometer (model: 10-AU, Turner Designs, Inc., San Jose, USA). Seawater samples (10 mL) were collected and used for nutrient analysis using a continuous segmented flow analyzer (model: QuAAtro 2-HR, BL TEC K.K., Tokyo, Japan)." (Page 7, Lines 23-27)

*- Figure 2: please also include the representation of the different periods P1-P5 at the top of the right section (figs. 2e, 2f, 2g) to better see the intervals of the values. And add in the caption of the figure, the abbreviations used in the graphs, so that reader doesn't need to look up the meaning every time in the text.*

We corrected as suggested in the revised manuscript.

*- Figure 3: when this figure is explained in the text (pag. 8, lines: 10-13), it is stated that figs. 3c and 3d also show Chl-a dispersion, but they only show the dispersion of TEP (3c) and CSP (3d). Please modify it. Also in the text, it's a bit confusing when the average values are mentioned, since the Chl-a values are included. In general, the structure of the paragraph is not very clear. Could it be possible to specify better the results, mentioning the correct figs and restructuring a bit the order of the values? For example, figs 3c and 3d are commented before than 3a and 3b.*

The order of the description has been revised: the explanation of Chl-a (Fig. 3a) is first, followed by the temporal variation of TEP and CSP (Fig. 3b) and the geographic distribution (Figs. 3c and 3d).

"Temporal variations of Chl-*a* and biological organic gel particles (TEPs and CSPs) in the surface seawater are shown in Fig. 3a and 3b. The concentration of Chl-*a* was high in the North Pacific Ocean and the Bering Sea (mean values ± one standard deviation ($1\sigma$): $0.86 \pm 0.23$ mg m$^{-3}$), coincident with high nutrient contents (Fig. S1), suggesting high biological activity (e.g., a phytoplankton bloom). The particular organic matter as TEPs and CSPs were also relatively high in the North Pacific Ocean and the Bering Sea (mean values ± $1\sigma$ : $73 \pm 34$ μg XGeq L$^{-1}$ and $24 \pm 22$ μg BSAeq L$^{-1}$, respectively), following the trend of Chl-*a*. The spatial distributions of TEPs and CSPs are also presented in Fig. 3c and 3d. Particularly, high concentrations of bioindicators (i.e., TEPs, CSPs) were observed from the Bering Sea to the Chukchi Sea (P2 and P4, Figs. 3c–d), corresponding to changes in nutrient concentrations. Conversely, over the Arctic Ocean, the concentration of CSPs decreased markedly to $12 \pm 13$ μg BSAeq L$^{-1}$, while TEPs and Chl-*a* maintained relatively high concentrations ($47 \pm 10$ μg XGeq L$^{-1}$ and $0.33 \pm 0.12$ mg m$^{-3}$, respectively) in comparison with those previously reported during summer (August–September) (Park et al., 2019, TEPs: ~20 μg XGeq L$^{-1}$, CSPs: ~20 μg BSAeq L$^{-1}$, and Chl-*a*: ~0.2 mg m$^{-3}$)." (Page 9, Lines 2-13)

*- Pag 10: Line 2: Could you specify a reference where the DNA staining method is conducted? Or did you do this analysis? If yes, could you include it in the results or supplementary section?*

The DNA-staining method described here is based on our previous study (Kawana et al., 2021)

and it was not obtained in this study.

"Our previous oceanographic observations over the central Pacific (Kawana et al., 2021) similarly showed that FAPs in Type A and Type C predominated (75 %) in clean remote oceanic air masses, and that their abundance correlated well with oceanic TEPs (polysaccharide polymer) and bacteria, when considering the influence of WS in the formation of SSAs, while FAPs in Type B were dominant (30 %) near land and strongly correlated with CSPs (protein-like polymers). The identity of marine bioaerosols detected by fluorescence observations was certified by comparison to a DNA staining method during our previous research cruise (Kawana et al., 2021)." (Page 10, Lines 21-27)

*- Page 11: The results obtained with CSPs are not discussed. Please add a brief summary.*
We added discussion for CSPs, in addition to TEPs. We added the following sentence.
"CSP-containing particles may fluoresce due to their amino-acid structure and may simultaneously exhibit ice nucleating ability (Hill et al., 2014; Fröhlich-Nowoisky et al., 2016). Considering the correlation between fluorescent bioaerosols and CSP were lower (0.5–0.6) in our previous study over the Central Pacific, our results here for the high latitudes represent a successful case where the high correlation coefficients between them were found, suggesting the necessity of ocean-specific studies in the future." (Page 12, Lines 6-10)

*- Section 3.4: Lines 2 and 3. Please make a revision of the text, since the commented values for the fig. 6 do not appear to be the same as those represented. In addition, the highest peaks are for P4 and P5 and not for P1 as indicated in the text.*
We believe that averages over log-normal distributions in the plots were correctly explained in the previous text. However, for conciseness we omitted the plots and corresponding texts in revision.

*- Figure 7: Graphs a) and e) represent the same data. Would it be possible to delete one of them? It is a bit confusing. Also, specify the meaning of the abbreviation "AS" in the figure caption.*
We would like to keep both figures to show overall temporal variation patterns in Fig. 6a (changed from Fig.7a) and their averages over the five periods segregated by air mass types in Fig. 6e (changed from Fig.7e). We added the meaning "AS" in the figure caption as suggested.

*- Section 3.5: Pag 15: it is sometimes difficult to follow in the text to which graphs the mentioned values refer. I would recommend adding the figure or table number in brackets after each explanation of the values in the text. Similar happens for figure 9 in page 16, it's not commented or specified the values. Furthermore, fig 9. shows all fluorescent particles together and does not distinguish between fine and coarse, whereas it can be deducted from the text that fig 9.*

*Shows them separately. Please, explain it better.*

In section 3.5, We have added to the description which values correspond to the figure and table number in brackets. Fig 8 (originally Fig.9) plots the INP number concentration against all fluorescent particles in the fine mode to generate the relationship equation. The coarse mode particles were not included here due to low correlation. We added the additional explanation in the caption of Fig.8.

"Scatterplot of total fluorescent fine particles (1 < Dp < 2.5 μm, sum of types A, B, C, AB, AC, BC, and ABC) and INP number concentrations measured at temperatures higher than −18, −24, and −30 °C. (Page 43)

*- A list of abbreviations is necessary.*

We added the list of abbreviations as an appendix in the revised manuscript.

*B. Minor changes:*

*• Abstract:*

*Last line of abstract, change "contributed" by "point", "define".*

We kept the word "contributed" because we thought it was more appropriate.

*• Introduction:*

*Pag 4:*

*- Line3: specify in brackets what bioindicators you go to analyse: TEP, CSP and Chla. If it's the first time that Chl-a is mentioned, also write the complete word.*

We corrected as suggested in the revised manuscript.

*• Results and Discussion:*

*Pag 7:*

*- Line 26: where are represented the plots of mass concentrations of OM and Na+ mentioned in the text? Specify it in parenthesis.*

This plot was not included in the manuscript because it was for reference information and we omitted this description in the revised manuscript.

*- Line 28: could you mention a brief explanation why the OC, sulfate and sea salt were found in the period P4?*

We added the following sentence.

"During P4, OC, sulfate, and sea salt were dominant (15, 31, and 43 %, respectively), suggesting marine influence with relatively high biological activity in this period."
(Page 8, Lines 19-20)

*Pag 8:*

*- Line 20: What do you mean by "in this area"? Do you mean the P2 and P4 periods? It is not clear.*

*- Line: 28: same as in line 20 when mentioning: "in this period".*

We added the following sentences, respectively.

"In the Bering Sea and the Chukchi Sea, …" (Page 9, Line 14)

"during P2, P3, and P4, ...." (Page 9, Line 23)

*Pag 14:*

*- Line 25: the word "elsewhere" can refer to literally another place. Please specify whether it refers to another place on land, marine...*

We corrected as suggested in the revised manuscript.

"pristine marine $N_{INP}$ values are approximately one order of magnitude lower than in the North Pacific Ocean" (Page 15, Lines 6-7)

*• Conclusions:*

*Pag 19: You explain the periods P1-P4, but you forgot to mention the P5. Could you add something about this period?*

There were a few descriptions for P5 because of the number of observation points were limited. But upon revision, we added some more results here together with P1 as the influence of Asian continental and terrestrial air masses was common.

"During P1 and P5 over the North Pacific Ocean, largely influenced by long-range transport from terrestrial regions including the Asian continent, particle chemical compositions were characterized by high mass fractions of Organics and sulfate (15–22 % and 28–48 %, respectively). In contrast, during P2, P3, and P4 over the Bering Sea and the Arctic Ocean, mainly influenced by maritime air masses, high mass fractions of sea salt (43–88 %) were noted." (Page 18, Line 28-Page 19, Line 3)

*C. Typographical corrections:*

*1. Introduction:*

*Pag.2:*

*- Line 11, change "and" by "or"*

We corrected as suggested in the revised manuscript.

*- Line 17: full stop after the parenthesis.*

We corrected as suggested in the revised manuscript.

*- Line: 19: close parenthesis after TEP; When you mention "microlayer" at the end of the line, which one do you mean, "microlayer of water"? Specify it.*

We changed the word "microlayer" as "sea surface microlayer" in the revised manuscript.

(Page 2, Line 20)

*Pag.3:*
*- Line: 6: change "biomaterials" by "biological organisms"*

We changed the word "biomaterials" to "microorganism and biological substances" in the revised manuscript. (Page 3, Line 10)

*- Line 21: in the Mediterranean Sea there is another study where INPs are analysed. It was carried out on an island in the Mediterranean Sea. Then, if you are referring to analyses carried out on a ship, don't mention it, but if you are referring to other analyses of bioparticles of marine origin where INP was analysed, mention that too.*
*The reference is: Tang, K., Sánchez-Parra, B., Yordanova, P., Wehking, J., Backes, A. T., Pickersgill, D. A., Maier, S., Sciare, J., Pöschl, U., Weber, B., and Fröhlich-Nowoisky, J.: Bioaerosols and atmospheric ice nuclei in a Mediterranean dryland: community changes related to rainfall, Biogeosciences, 19, 71–91, https://doi.org/10.5194/bg-19-71-2022, 2022*

We added a reference (Tang et al., 2022) in the revised manuscript.

*Pag 4:*
*- Line 1: add a "s" to word "cycle".*

We corrected as suggested in the revised manuscript.

*Pag 10:*
*- Line 16: Rewrite it, specifying that the high correlation appears for "most" of bioindicators, or mention that" for TEP is lower than for CSP and Chl-a".*

Although examined, we could not catch the point of the reviewer's comment. We believe that this part adequately described correlation coefficients between bioaerosols and sea-surface biological parameters.

*- Lines 24 and 25: "the correlation coefficients were almost unchanged when an exponent of 2 was used instead of 1 (see Table 2)". Do you mean between which values? Be a little more specific.*

We corrected explanations in the revised manuscript as follows:

"when the wind effect was considered as the square of WS ($WS^2$), high correlation coefficients were also obtained (Table 2, *R*: 0.87–0.90)." (Page 11, Lines 19-20)

*2. Cruise Observations:*
*Pag: 4*
*-Line: 9: add a "s" to "measurement"*

We corrected as suggested in the revised manuscript.

*specify what is dp and what is referred as "dry" and "wet" in the equation.*

We added the following sentences:

"In the calculation of $\kappa_{CCNC}$, the $d_{act}$ was applied as $d_{p,dry}$ and the particle diameter in humidified conditions was assumed as $d_{p,wet}$, and the $S$ was applied as the SS conditions in the observation." (Page 5, Lines 23-24)

*Pag 7*

*-Line 5: delete the brackets after "respectively".*

We corrected as suggested in the revised manuscript.

*3. Results and discussion:*

*Pag 9:*

*- Line 11: at the end of the line is said that "…representing 2.5%" Is this value correct? Wouldn't you mean 25%? If it's correct, in the line 20, is the value 7% correct too? Please, revise it.*

We doublechecked that the original values (2.5% and 7%, respectively) are correct.

*- Line 15: add after "marine biogenic source" the following: "(according to our bioindicators analysis)".*

We corrected as suggested in the revised manuscript.

*Pag 10:*

*- Line 15: add "fluorescent" before the word "particles"*

We corrected as suggested in the revised manuscript.

*- Line 26: change "marine" by "fluorescent" or add "fluorescent" between "marine" and "bioaerosols".*

We corrected as suggested in the revised manuscript.

*Pag 11:*

*- Line 8: delete "(with bacteria, in some cases)". You don't have evidence of this in the present study.*

We corrected as suggested in the revised manuscript.

*- Line 15: change "formation" by "detection"*

We corrected the word "formation" as "evolution" in the revised manuscript.

*- Line 29: change "study" by "studies"*

We corrected as suggested in the revised manuscript.

*Pag 12:*
*- Line 15: it seems that there is an extra space between "Ocean" and "during". Delete the space.*
We deleted the space as suggested in the revised manuscript.

*Pag 14:*
*- Line 19: change Fig. "8d" by "7d"*
*- Line 20: change Fig "7e" by "7d"*
We corrected as suggested in the revised manuscript.
We now numbered as Figs.6d, 6e, and 6f, because the original Fig 6 has been deleted in the revised manuscript.

*Pag 15:*
*- Line 11: change Table "2" by "3"*
We corrected as suggested in the revised manuscript.

*- Line 26: after "…and strong winds" make a full stop.*
We corrected as suggested in the revised manuscript.

*4. Conclusions:*
*Pag 19, Line 9: delete "i.e" as you only analysed the three mentioned bioindicators (TEP, CSP and Chl-a) and no more.*
We kept the word "i.e."; there might be confusion with "e.g."

---

## Author Comment (AC2)

*Reviewer #2:*

*General comment*

*This study evaluated if the marine biological activity present in the North Pacific Ocean, Bering Sea, and Arctic Ocean can impact biological aerosol particles formation, and the distribution of CCNs and INPs. To answer this, a large set of measurements were performed between September 27 and November 10, 2019 in an oceanographic cruise. The author found that bioindicators, fluorescent particles, INP, and CCN correlate positively, indicating that marine biota contribute substantially as a source of bioaerosols and cloud formation via INP and CCN over the remote Arctic Ocean during periods of high biological activity. The results are of high importance for this remote, valuable, and uncertain region. The measurements were properly performed and the data analyses are correct and consistent. Although the is well-written several parts can be improved. The present manuscript can be accepted for its publication in ACP after the following comments are properly addressed.*

We would like to thank the reviewer for providing valuable comments and insights on our work. Our responses are given below. All changes are shown in color in the revised manuscript.

*Major comments:*

*1. All the correlations indicated along the manuscript need to be defined as statistically significant or not.*

We have added the P-values for data with correlations larger than 0.5, to show statistical significance in Tables 2 and 3.

*2. The discussion of the present results with literature data needs to be improved avoiding redundancy and trying to be concise and clear.*

We removed figures and texts about aerosol number concentration and number-size distribution obtained by SMPS, to focus on the CCN and INPs discussions. Overall, the discussion of relationship between bioindicators derived from marine ecosystems, their correlation with fluorescent bioaerosols and particles activated as CCN and INP has been revised to be more concise, particularly in Section 3.4.

*3. Some figures need to be improved.*

We have carefully reviewed and made revisions for clarity.

*Minor comments:*

*P1, L23: Why the CCN concentration for the Arctic Ocean is indicated as a single value (36 cm$^{-3}$) and not as a range?*

For the characterization with air mass, we have classified periods dominated by terrestrial and Asian continental influences (P1 and P5), periods dominated by marine and biogenic sources (P2 and P4), and periods in the Arctic Ocean (P5). The averaged $N_{CCN}$ in the Arctic

Ocean is described as a single value because it is derived only from a single P3 period. See Table 1 for full ranges and standard deviations.

*P1, L26-27: "The averaged INP concentration (NINP) measured at temperatures of −18 and −24 °C with marine sources was 0.01–0.09 and 0.1–2 L−1, respectively" Please indicate the region or period you are referring to.*

We corrected in the revised manuscript.

"The averaged INP concentration ($N_{INP}$) measured at temperatures of −18 and −24 °C with marine sources in the North Pacific and Bering Sea was 0.01–0.09 and 0.1–2.5 L$^{-1}$, respectively, and that over the Arctic Ocean was 0.001–0.016 and 0.012–0.27 L$^{-1}$, respectively." (Page 1, Lines 28-30)

*P3, L6: I am not sure that the term "biomaterials" is appropriate. Please reconsider this.*

We changed the word "biomaterials" to "microorganism and biological substances" in the revised manuscript. (Page 3, Line 10)

*P6, L20: I am not sure that the term "number densities of INPs" is appropriate. Please reconsider this.*

We changed the word "number densities of INPs" to "number concentrations of INPs" (Page 6, Lines 24)

*P7, L15: should "P3, 11–27 October 2019" be "P3, 10–27 October 2019" as indicated in Table 1?*

Yes, exactly. We corrected as suggested in the revised manuscript.

*P8, L20, L26 and a long the text: I think it would be more appropriate to call "bloom" and "autumn bloom" as "phytoplankton bloom".*

We changed the words "bloom" and "autumn bloom" to "phytoplankton bloom" in the revised manuscript.

*P11, L2-3: "suggesting the possibility of dependence on phytoplankton communities in the different oceanic regions". What is reported in the Literature?*

We added the reference (Taylor et al., 2014) in the revised manuscript.

*P12: In line 13 "Previous studies" is mentioned; however, only Park et al. (2020) is cited. Please add more studies.*

This description is removed because we have excluded a figure and discussion on the aerosol number-size distribution and aerosol concentrations in the revised manuscript.

*P13, L6: I think "such as dicarboxylic acid (i.e., oxalic acid)" should be "such as oxalic acid".*

We corrected as suggested in the revised manuscript.

*Section 3.4. The discussion related to CCNs is very confused and hard to follow. Please improve this part.*

In Section 3.4, we have excluded data and discussion on the aerosol number concentrations and number-size distribution in the revised manuscript to forces on the CCN and INPs discussions. We also revised the description of CCN parameterization briefly.

*P15, L14: "INP formation". Aerosol particles can form but not an INP. An aerosol particle is capable or not to act as INP.*

We changed the word "INP formation" as "INP activation" in the revised manuscript.
(Page 15, Line 13)

*P17, L3: Why κ value for the accumulation mode is indicated as a single value (0.57) and not as a range?.*

This description is based on the literature (Gong et al., 2019), not our results, but κ value for the accumulation mode is single parameter because the CCN activation diameter in the accumulation mode is obtained at 0.08% SS only.

*P17, L23-29: I have the impression that this paragraph is repetitive.*

We corrected in the revised manuscript.
"It is also unique to find that simultaneously observed NCCN and NINP showed positive correlations (R: 0.95–0.97, Table 3), suggesting that enhanced marine biological sources and the formed fluorescent marine bioaerosols contribute to the increases of both CCN and INPs during a phytoplankton bloom, despite the reduced CCN activity due to the less hygroscopicity. This is consistent with the results from microcosm experiments, which suggest that OM coating on SSAs reduces the surface tension, enhances the emission flux of the particle number concentration, and thus the CCN concentration (Alpert et al., 2015; Ito et al., 2023)." (Page 17, Lines 14-20)

*P18, L20: Why averaged NCCN at 0.4 % SS for the Arctic Ocean is indicated as a single value (36 cm−3) and not as a range?*

For the characterization with air mass, we have classified periods dominated by terrestrial and Asian continental influences (P1 and P5), periods dominated by marine and biogenic sources (P2 and P4), and periods in the Arctic Ocean (P5). The averaged $N_{CCN}$ in the Arctic Ocean is described as a single value because it is derived only from the P3 period. See Table 1 for full ranges and standard deviations.

*Tables 2 and 3: indicate if the correlations are statistically significant or not.*

We have added the P-values for data with correlations larger than 0.5, to show statistical significance in Tables 2 and 3.

*Figure 1: The can different periods P1 to P5 be distinguished in the cruise track?*

The cruise track during the P1 and P5 periods overlapped but the segments can be recognized from trajectory start points in each panel. We believe that this is clearly depicted in Figure 1 with separate panels.

*Figure 2: Add labels to panel a, b, c, and d. Also, change the color for EC and NO3 as they look very similar.*

We corrected as suggested in the revised manuscript.

*Figure 4: I am not sure if panels c and d can be more useful with scatter plots?*

The plot of fluorescent particles versus wind speed is presented and discussed in Fig. 5 and here we would like to keep Fig. 4c and Fig. 4d to check the response of the fluorescent particle concentrations to the local wind speed in time series.

*Figure 7: y-axis in panel f should go in log scale and y-axis in panel (f) should read "INP concentration".*

We corrected as suggested in the revised manuscript (now Fig. 6f).

*Technical comments:*

*P2, L5: Add a reference after "processes".*

We added a reference (Ramanathan et al., 2001) in the revised manuscript.

*P2, L10: Add a reference after "(INPs)".*

We added a reference (Brooks and Thornton, 2018) in the revised manuscript.

*P2, L19: "(TEP and protein-containing Coomassie stainable particle (CSP)" It seems that a bracket is missing.*

We corrected it in the revised manuscript.

*P3, L1: Add a reference after "INPs".*

We added references (Sun and Ariya, 2006; Murray et al., 2012) in the revised manuscript.

*P3, L3-4: "Some CCN grow to giant CCN". Is it not possible to have a primary GCCN?*

We corrected this sentence as follows:

"Some particles may form giant CCN and be activated as cloud droplets of larger size" (Page

3, Lines 7-8).

We added a reference (Xie et al., 2021) in the revised manuscript.

We added a reference (Furutani et al., 2008) in the revised manuscript.

*P8, L13: I think "…contents and suggesting…" should be "…contents, suggesting…"*
We corrected as suggested in the revised manuscript.

*P9, L12: "total" of what?*
*P9 L20: "fraction" of what?*
We corrected this sentence as follows:
"representing a 2.5 % fraction of the total particles." (Page 10, Lines 5-6)
"the fraction of fluorescent particles in the total particles was 7 %." (Page 10, Line 14)

We added the references (Russell et al., 2010; Engel et al., 2017; Park et al., 2019) in the revised manuscript.

*P11, L29: "study" should be "studies".*
We corrected as suggested in the revised manuscript.

We added the references (Facchini et al., 2008; Ovadnevaite et al., 2011) in the revised manuscript.

*P13, L32 to P14, L4: Please improve the punctuation here. This is a very long paragraph.*
We corrected this sentence as follows:
"CCN activation during summer was affected largely by the less hygroscopic OM associated with marine and terrestrial biological activity, while CCN activation during winter was mainly determined by highly hygroscopic components such as sea salt with limited biological activity (Kawana et al., 2022a)." (Page 14, Lines 15-17)

*P14, L19: "(Fig. 8d)" should be "(Fig. 7d)"*
We corrected as suggested in the revised manuscript.
We now numbered as Figs.6d because the original Fig 6 has been deleted in the revised manuscript.

*P14, L25: I think "lower than elsewhere (i.e., North Pacific Ocean)." Should be "lower than in the North Pacific Ocean."*

We corrected as suggested in the revised manuscript.

*P14, L30: Add a reference after "INPs".*

We added a reference (Welti et al., 2020) in the revised manuscript.

*P15, L17: I think "but still significant" should be "but still strong" or something similar.*

We changed the word "still significant" as "still high" in the revised manuscript. (Page 15, Line 29)

*Figure 4: I think "(black line) and fluorescent" should be "(black line), fluorescent"*

We corrected as suggested in the revised manuscript.

*Reference:*

Alpert, P. A., W. P. Kilthau, D. W. Bothe, J. C. Radway, J. Y. Aller and D. A. Knopf.: The influence of marine microbial activities on aerosol production: a laboratory mesocosm study, J. Geophys. Res.: Atmos., 120, 8841–8860, DOI: 10.1002/2015JD023469, 2015.

Brooks, S. D., and D. C. O. Thornton: Marine aerosols and clouds, Annu. Rev. Mar. Sci. 2018, 10, 289-313, https://doi.org/10.1146/annurev-marine-121916-063148.

Engel, A., J. Piontek, K. Metfies, S. Endres, P. Sprong, I. Peeken, S. Gäbler-Schwarz, E-M Nöthig.: Inter-annual variability of transparent exopolymer particles in the Arctic Ocean reveals high sensitivity to ecosystem changes, Sci. Rep., 7, 4129, doi: 10.1038/s41598-017-04106-9, 2017.

Facchini, M. C., Rinaldi, M., Decesari, S., Carbone, C., Finessi, E., Mircea, M., et al.: Primary submicron marine aerosol dominated by insoluble organic colloids and aggregates, Geophysical Research Letters, 35, L17814, 2008.

Murray, B. J., D. O'Sullivan, J. D. Atkinson, M. E. Webb.: Ice nucleation by particles immersed in supercooled cloud droplets, Chem. Soc. Rev., 41, 6519-6554, 2012.

Ovadnevaite, J., Ceburnis, D., Martucci, G., Bialek, J., Monahan, C., Renaldi, M., et al.: Primary marine organic aerosol: A dichotomy of low hygroscopicity and high CCN activity, Geophysical Research Letters, 38, L21806, 2011.

Park, J., M. Dall'Osto, K. Park, J-H. Kim, J. Park, K-T. Park, C. Y. Hwang, G. I. Jang, Y. Gim, S. Kang, S. Park, Y. K. Jin, S. S. Yum, R. Simo, Y. J. Yoon.: Arctic primary aerosol production strongly influenced by riverine organic matter, Environ. Sci. Technol., 53, 8621-8630, doi: 10.1021/acs.est.9b03399, 2019.

Ramanathan, V., P. J. Crutzen, J. T. Kiehl, and D. Rosenfeld.: Atmosphere—Aerosols, climate, and the hydrological cycle, Science, 294, 5549, 2119–2124, 2001.

Sun, J., and P. A. Ariya.: Atmospheric organic and bio-aerosols as cloud condensation nuclei (CCN): A review, Atmos. Environ., 40, 5, 795-820, doi: 10.1016/j,atmosenv.2005.05.052, 2006.

Taketani, F., Miyakawa, T., Takigawa, M., Yamaguchi, M., Komazaki, Y., Mordovskoi, P., Takashima, H., Zhu, C., Nishino, S., Tohjima, Y., and Kanaya, Y.: Characteristics of atmospheric black carbon and other aerosol particles over the Arctic Ocean in early autumn 2016: Influence from biomass burning as assessed with observed microphysical properties and model simulations, Science of the Total Environment, 848, 157671, http://dx.doi.org/10.1016/j.scitotenv.2022.157671, 2022.

Taylor, J., Cottingham, S., Billinge, J. et al.: Seasonal microbial community dynamics correlate with phytoplankton-derived polysaccharides in surface coastal waters. ISME J 8, 245–248, https://doi.org/10.1038/ismej.2013.178, 2014.

Tobo, Y.: An improved approach for measuring immersion freezing in large droplets over a wide temperature range, Sci. Rep., 6, 32930, doi: 10.1038/srep32930, 2016.

Tobo, Y., Uetake, J., Matsui, H., Moteki, N., Uji, Y., Iwamoto, Y., et al.: Seasonal trends of atmospheric ice nucleating particles over Tokyo. Journal of Geophysical Research: Atmospheres, 125, e2020JD033658. https://doi.org/10.1029/2020JD033658, 2020.

Welti, A., Bigg, E. K., DeMott, P. J., Gong, X., Hartmann, M., Harvey, M., Henning, S., Herenz, P., Hill, T. C. J., Hornblow, B., Leck, C., Löffler, M., McCluskey, C. S., Rauker, A. M., Schmale, J., Tatzelt, C., van Pinxteren, M., and Stratmann, F.: Ship-based measurements of ice nuclei concentrations over the Arctic, Atlantic, Pacific and Southern oceans, Atmos. Chem. Phys., 20, 15191–15206, https://doi.org/10.5194/acp-20-15191-2020, 2020.

Xie, W., Li, Y., Bai, W., Hou, J., Ma, T., Zeng, X., Zhang, L., An, T.: The source and transport of bioaerosols in the air: A review. Front. Environ. Sci. Eng., 15, 3: 44 https://doi.org/10.1007/s11783-020-1336-8, 2021.